# Domain constraints improve risk prediction when outcome data is missing

**Sidhika Balachandar** *
Cornell Tech

**Nikhil Garg**
Cornell Tech

**Emma Pierson**
Cornell Tech

## Abstract

Machine learning models are often trained to predict the outcome resulting from a human decision. For example, if a doctor decides to test a patient for disease, will the patient test positive? A challenge is that historical decision-making determines whether the outcome is observed: we only observe test outcomes for patients doctors historically tested. Untested patients, for whom outcomes are unobserved, may differ from tested patients along observed and unobserved dimensions. We propose a Bayesian model class which captures this setting. The purpose of the model is to accurately estimate risk for both tested and untested patients. Estimating this model is challenging due to the wide range of possibilities for untested patients. To address this, we propose two domain constraints which are plausible in health settings: a *prevalence constraint*, where the overall disease prevalence is known, and an *expertise constraint*, where the human decision-maker deviates from purely risk-based decision-making only along a constrained feature set. We show theoretically and on synthetic data that domain constraints improve parameter inference. We apply our model to a case study of cancer risk prediction, showing that the model's inferred risk predicts cancer diagnoses, its inferred testing policy captures known public health policies, and it can identify suboptimalities in test allocation. Though our case study is in healthcare, our analysis reveals a general class of domain constraints which can improve model estimation in many settings.

## 1 Introduction

Machine learning models are often trained to predict outcomes in settings where a human makes a high-stakes decision. In criminal justice, a judge decides whether to release a defendant prior to trial, and models are trained to predict whether the defendant will fail to appear or commit a crime if released (Lakkaraju et al., 2017; Jung et al., 2020a; Kleinberg et al., 2018). In lending, a creditor decides whether to grant an applicant a loan, and models are trained to predict whether the applicant will repay (Björkegren & Grissen, 2020; Crook & Banasik, 2004). In healthcare—the setting we focus on in this paper—a doctor decides whether to test a patient for disease, and models are trained to predict whether the patient will test positive (Jehi et al., 2020; McDonald et al., 2021; Mullainathan & Obermeyer, 2022). Machine learning predictions help guide decision-making in all these settings. A model which predicts a patient's risk of disease can help allocate tests to the highest-risk patients, and also identify suboptimalities in human decision-making: for example, testing patients at low risk of disease, or failing to test high risk patients (Mullainathan & Obermeyer, 2022).

A fundamental challenge in all these settings is that historical decision-making determines whether the outcome is observed. In criminal justice, release outcomes are only observed for defendants judges have historically released. In lending, loan repayments are only observed for applicants historically granted loans. In healthcare, test outcomes are only observed for patients doctors have historically tested. This is problematic because the model must make accurate predictions for the entire population, not just the historically tested population. Learning only from the tested population also risks introducing bias against underserved populations who are less likely to get medical tests partly due to worse healthcare access (Chen et al., 2021; Pierson, 2020; Servik, 2020; Jain et al., 2023). Thus, there is a challenging distribution shift between the tested and untested populations. The

---

*Correspondence to: sidhikab@cs.cornell.edu

two populations may differ both along *observables* recorded in the data and *unobservables* known to the human decision-maker but unrecorded in the data. For example, tested patients may have more symptoms recorded than untested patients—but they may also differ on unobservables, like how much pain they are in or how sick they look, which are known to the doctor but are not available for the model. This setting, referred to as the *selective labels* setting (Lakkaraju et al., 2017), occurs in high-stakes domains including healthcare, hiring, insurance, lending, education, welfare services, government inspections, tax auditing, recommender systems, wildlife protection, and criminal justice and has been the subject of substantial academic interest (see §6 for related work).

Without further constraints on the data generating process, there is a wide range of possibilities for the untested patients. They could all have the disease or never have the disease. However, selective labels settings often have *domain-specific constraints* which would allow us to limit the range of possibilities. For example, in medical settings, we might know the prevalence of a disease in the population. Recent distribution shift literature has shown that generic methods generally do not perform well across all distribution shifts and that domain-specific constraints can improve generalization (Gulrajani & Lopez-Paz, 2021; Koh et al., 2021; Sagawa et al., 2022; Gao et al., 2023; Kaur et al., 2022; Tellez et al., 2019; Wiles et al., 2022). This suggests the utility of domain constraints in improving generalization from the tested to untested population.

Motivated by this reasoning, we make the following contributions:

1. We propose a Bayesian model class which captures the selective labels setting and nests classic econometric models. We model a patient's risk of disease as a function of observables and unobservables. The probability of testing a patient increases with disease risk and other factors (e.g., bias). The purpose of the model is to accurately estimate risk for both the tested and untested patients and to quantify deviations from purely risk-based test allocation.

2. We propose two constraints informed by the medical domain to improve model estimation: a *prevalence constraint*, where disease prevalence is known, and an *expertise constraint*, where the decision-maker deviates from risk-based decision-making along a constrained feature set. We show theoretically and on synthetic data that the constraints improve inference.

3. We apply our model to a breast cancer risk prediction case study. We conduct a suite of validations, showing that the model's (i) inferred risks predict cancer diagnoses, (ii) inferred unobservables correlate with known unobservables, (iii) inferred predictors of cancer risk correlate with known predictors, and (iv) inferred testing policy correlates with public health policies. We also show that our model identifies deviations from risk-based test allocation and that the prevalence constraint increases the plausibility of inferences.

Though our case study is in healthcare, our analysis reveals a general class of domain constraints which can improve model estimation in many selective labels settings.

## 2 MODEL

We now describe our Bayesian model class. Following previous work (Mullainathan & Obermeyer, 2022), our underlying assumption is that whether a patient is tested for a disease should be determined primarily by their risk of disease. Thus, the purpose of the model is to accurately estimate risk for both the tested and untested patients and to quantify deviations from purely risk-based test allocation. The latter task relates to literature on diagnosing factors affecting human decision-making (Mullainathan & Obermeyer, 2022; Zamfirescu-Pereira et al., 2022; Jung et al., 2018).

Consider a set of people indexed by $i$. For each person, we see observed features $X_i \in \mathbb{R}^D$ (e.g., demographics and symptoms in an electronic health record). We observe a *testing decision* $T_i \in \{0, 1\}$, where $T_i = 1$ indicates that the $i$th person was tested. If the person was tested ($T_i = 1$), we observe an outcome $Y_i$. $Y_i$ might be a binary indicator (e.g. $Y_i = 1$ means that the person tests positive), or $Y_i$ might be a numeric outcome of a medical test (e.g. T cell count or oxygen saturation levels). Throughout, we generally refer to $Y_i$ as a binary indicator, but our framework extends to non-binary $Y_i$, and we derive our theoretical results in this setting. If $T_i = 0$ we do not observe $Y_i$.

There are *unobservables* (Angrist & Pischke, 2009; Rambachan et al., 2022), denoted by $Z_i \in \mathbb{R}$, that affect *both* $T_i$ and $Y_i$ but are not recorded in the dataset – e.g., whether the doctor observes that the

person is in pain. Consequently, the risk of the tested population differs from the untested population even conditional on observables $X_i$: i.e. $p(Y_i|T_i = 1, X_i) \neq p(Y_i|T_i = 0, X_i)$.

A person's risk of disease is captured by their *risk score* $r_i \in \mathbb{R}$, which is a function of $X_i$ and $Z_i$. Whether the person is tested ($T_i = 1$) depends on their risk score $r_i$, but also factors like screening policies or socioeconomic disparities. More formally, our data generating process is

$$
\begin{aligned}
\text{Unobservables:} \quad & Z_i \sim f(\cdot|\sigma^2) \\
\text{Risk score:} \quad & r_i = X_i^T \boldsymbol{\beta_Y} + Z_i \\
\text{Test outcome:} \quad & Y_i \sim h_Y(\cdot|r_i) \\
\text{Testing decision:} \quad & T_i \sim h_T(\cdot|\alpha r_i + X_i^T \boldsymbol{\beta_\Delta}).
\end{aligned}
\tag{1}
$$

In words, $Z_i$ is drawn from a distribution $f$ with parameter $\sigma^2$, which captures the relative importance of the unobserved versus observed features. The disease risk score $r_i \in \mathbb{R}$ is modeled as a linear function of observed features (with unknown coefficients $\boldsymbol{\beta_Y} \in \mathbb{R}^D$) and the unobserved $Z_i$. $Y_i$ is drawn from a distribution $h_Y$ parameterized by $r_i$ – e.g., $Y_i \sim \text{Bernoulli}(\text{sigmoid}(r_i))$. Analogously, the testing decision $T_i$ is drawn from a distribution $h_T$ parameterized by a linear function of the true disease risk score and other factors, with unknown coefficients $\alpha \in \mathbb{R}$ and $\boldsymbol{\beta_\Delta} \in \mathbb{R}^D$. Because $T_i$ depends on $r_i$, and $r_i$ is a function of $Z_i$, $T_i$ depends on $Z_i$. Figure 1 illustrates the effect of $\alpha$ and $\boldsymbol{\beta_\Delta}$. A larger $\alpha$ indicates that testing probability increases more steeply in risk. $\boldsymbol{\beta_\Delta}$ captures human or policy factors which affect a patient's probability of being tested beyond disease risk. In other words, $\boldsymbol{\beta_\Delta}$ captures deviations from purely risk-based test allocation. Putting things together, the model parameters are $\theta \triangleq (\alpha, \sigma^2, \boldsymbol{\beta_\Delta}, \boldsymbol{\beta_Y})$.

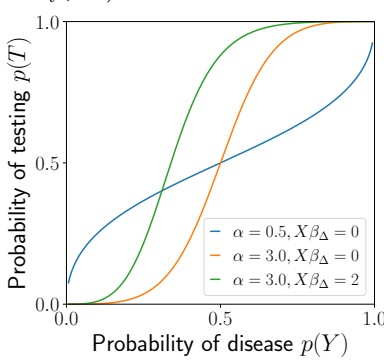

Figure 1: Effect of $\alpha$ and $X\boldsymbol{\beta_\Delta}$: $\alpha$ controls how steeply testing probability $p(T_i)$ increases in disease risk $p(Y_i)$, while $X\boldsymbol{\beta_\Delta}$ captures factors which affect $p(T_i)$ when controlling for $p(Y_i)$.

**Medical domain knowledge:** Besides the observed data, in medical settings we often have constraints to aid model estimation. We consider two constraints.

- **Prevalence constraint:** The average value of $Y$ across the entire population is known ($\mathbb{E}[Y]$). When $Y$ is a binary indicator of whether a patient has a disease, this corresponds to assuming that the *disease prevalence* is known. This assumption is plausible because estimating prevalence has been the focus of substantial public health research, and estimates thus exist in many medical settings; for more details see appendix A. For example, this information is available for cancer (Cancer Research UK), COVID-19 (NIH National Cancer Institute, 2023), and heart disease (CDC, 2007). In some cases, the prevalence is only *approximately* known (Manski & Molinari, 2021; Manski, 2020; Mullahy et al., 2021); our Bayesian formulation can incorporate such soft constraints as well.

- **Expertise constraint:** Because doctors and patients are informed decision-makers, we can assume that tests are allocated *mostly* based on disease risk. Specifically, we assume that there are some features which do not affect a patient's probability of receiving a test when controlling for their risk: i.e., that $\boldsymbol{\beta_\Delta}_d = 0$, for at least one dimension $d$. For example, we may assume that when controlling for disease risk, a patient's height does not affect their probability of being tested for cancer, and thus $\boldsymbol{\beta_\Delta}_{\text{height}} = 0$.

## 3 THEORETICAL ANALYSIS

In this section, we prove why our proposed constraints improve parameter inference by analyzing a special case of our general model in equation 1. In Proposition 3.1, we show that this special case is equivalent to the Heckman model (Heckman, 1976; 1979), which is used to correct bias from non-randomly selected samples. In Proposition 3.2, we analyze this model to show that constraints can improve the precision of parameter inference. The full proofs are in Appendix B. In Sections 4 and 5 we empirically generalize our theoretical results beyond the special Heckman case.

### 3.1 Domain constraints can improve the precision of parameter inference

We start by defining the Heckman model and showing it is a special case of our general model.

**Definition 1** (Heckman correction model). *The Heckman model can be written in the following form (Hicks, 2021):*

$$
\begin{aligned}
T_i &= \mathbb{1}[X_i^T \tilde{\boldsymbol{\beta}}_{\boldsymbol{T}} + u_i > 0] \\
Y_i &= X_i^T \tilde{\boldsymbol{\beta}}_{\boldsymbol{Y}} + Z_i \\
\begin{bmatrix} u_i \\ Z_i \end{bmatrix} &\sim Normal\left( \begin{bmatrix} 0 \\ 0 \end{bmatrix}, \begin{bmatrix} 1 & \tilde{\rho} \\ \tilde{\rho} & \tilde{\sigma}^2 \end{bmatrix} \right).
\end{aligned}
\tag{2}
$$

**Proposition 3.1.** *The Heckman model (Definition 1) is equivalent to the following special case of the general model in equation 1:*

$$
\begin{aligned}
Z_i &\sim \mathcal{N}(0, \sigma^2) \\
r_i &= X_i^T \boldsymbol{\beta}_{\boldsymbol{Y}} + Z_i \\
Y_i &= r_i \\
T_i &\sim Bernoulli(\Phi(\alpha r_i + X_i^T \boldsymbol{\beta}_{\boldsymbol{\Delta}})).
\end{aligned}
\tag{3}
$$

It is known that the Heckman model is identifiable (Lewbel, 2019), and thus the special case of our model is identifiable (i.e., distinct parameter sets correspond to distinct observed expectations) without further constraints. However, past work has often placed constraints on the Heckman model (though different constraints from those we propose) to improve parameter inference. Without constraints, the model is only weakly identified by functional form assumptions (Lewbel, 2019). This suggests that our proposed constraints could also improve model estimation. In Proposition 3.2, we make this intuition precise by showing that our proposed constraints improve the *precision* of the parameter estimates as measured by the *variance* of the parameter posteriors.

In our Bayesian formulation, we estimate a posterior distribution for parameter $\theta$ given the observed data: $g(\theta) \triangleq p(\theta|X, T, Y)$. Let $Var(\theta)$ denote the variance of $g(\theta)$. We show that constraining the value of any one parameter *will not worsen* the precision with which other parameters are inferred. In particular, constraining a parameter $\theta_{con}$ to a value drawn from its posterior distribution will not in expectation increase the posterior variance of any other unconstrained parameters $\theta_{unc}$. To formalize this, we define the *expected conditional variance*:

**Definition 2** (Expected conditional variance). *Let the distribution over model parameters $g(\theta) \triangleq p(\theta|X, T, Y)$ be the posterior distribution of the parameters $\theta$ given the observed data $\{X, T, Y\}$. We define the expected conditional variance of an unconstrained parameter $\theta_{unc}$, conditioned on the value of a constrained parameter $\theta_{con}$, to be $\mathbb{E}[Var(\theta_{unc}|\theta_{con})] \triangleq \mathbb{E}_{\theta_{con}^* \sim g}[Var(\theta_{unc}|\theta_{con} = \theta_{con}^*)]$.*

**Proposition 3.2.** *In expectation, constraining the parameter $\theta_{con}$ does not increase the variance of any other parameter $\theta_{unc}$. In other words, $\mathbb{E}[Var(\theta_{unc}|\theta_{con})] \leq Var(\theta_{unc})$. Moreover, the inequality is strict as long as $\mathbb{E}[\theta_{unc}|\theta_{con}]$ is non-constant in $\theta_{con}$ (i.e., $Var(\mathbb{E}[\theta_{unc}|\theta_{con}]) > 0$).*

In other words, we reason about the effects of fixing a parameter $\theta_{con}$ to its true value $\theta_{con}^*$. That value $\theta_{con}^*$ is distributed according to the posterior distribution $g$, and so we reason about expectations over $g$. In expectation, fixing the value of $\theta_{con}$ does not increase the variance of any other parameter $\theta_{unc}$, and strictly reduces it as long as the expectation of $\theta_{unc}$ is non-constant in $\theta_{con}$.

Both the expertise and prevalence constraints fix the value of at least one parameter. The expertise constraint fixes the value of $\boldsymbol{\beta}_{\boldsymbol{\Delta} d}$ for some $d$. For the Heckman model, the prevalence constraint fixes the value of the intercept $\boldsymbol{\beta}_{\boldsymbol{Y} 0}$ (assuming the standard condition that columns of $X$ are zero-mean except for an intercept column of ones). Thus, Proposition 3.2 implies that both constraints will not increase the variance of other model parameters, and will strictly reduce it as long as the posterior expectations of the unconstrained parameters are non-constant in the constrained parameters. In Appendix B we prove Proposition 3.2 and provide conditions under which the constraints strictly reduce the variance of other model parameters. We also verify and extend these theoretical results on synthetic data (Appendix D.1 Figure S1).

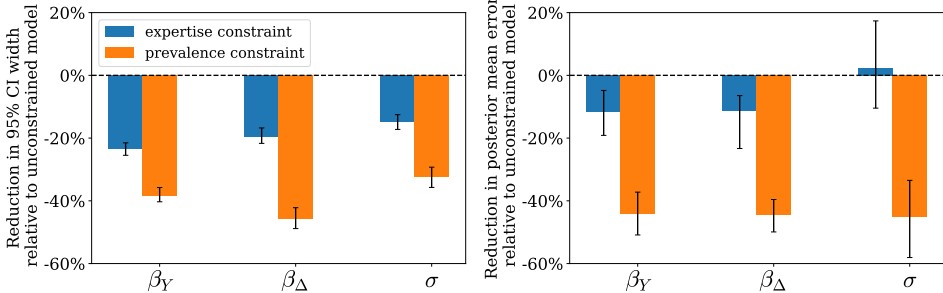

Figure 2: The prevalence and expertise constraints each produce more precise and accurate inferences on synthetic data drawn from the Bernoulli-sigmoid model with uniform noise (equation 4). To quantify precision (left), we report the percent reduction in 95% confidence interval width as compared to the unconstrained model. To quantify accuracy (right), we report the percent reduction in posterior mean error — i.e., the absolute difference between the posterior mean and the true parameter value — as compared to the unconstrained model. We plot the median across 200 synthetic datasets. Error bars denote the bootstrapped 95% confidence interval on the median.

## 3.2 EMPIRICAL EXTENSION BEYOND THE HECKMAN SPECIAL CASE

While we derive our theoretical results for a special case of our general model class, in our experiments (§4 and §5) we validate they hold beyond this special case by using a Bernoulli-sigmoid model:

$$
\begin{aligned}
Z_i &\sim \text{Uniform}(0, \sigma^2) \\
r_i &= X_i^T \boldsymbol{\beta_Y} + Z_i \\
Y_i &\sim \text{Bernoulli}(\text{sigmoid}(r_i)) \\
T_i &\sim \text{Bernoulli}(\text{sigmoid}(\alpha r_i + X_i^T \boldsymbol{\beta_\Delta})) \,.
\end{aligned}
\tag{4}
$$

We note two ways in which this model differs from the Heckman model. First, it uses a *binary* disease outcome $Y$ because this is an appropriate choice for our breast cancer case study (§5). With a binary outcome, models are known to be more challenging to fit: one cannot simultaneously estimate $\alpha$ and $\sigma$, and models fit without constraints may fail to recover the correct parameters (StataCorp, 2023; Van de Ven & Van Praag, 1981; Toomet & Henningsen, 2008). Even in this more challenging case, we show that our proposed constraints improve model estimation. Second, this model uses a uniform distribution of unobservables instead of a normal distribution of unobservables. As we show in Appendix C, this choice allows us to marginalize out $Z_i$, greatly accelerating model-fitting.

## 4 SYNTHETIC EXPERIMENTS

We now validate our proposed approach on synthetic data. Our theoretical results imply that our proposed constraints should reduce the variance of parameter posteriors (improving precision). We verify that this is the case. We also show empirically that the proposed constraints produce posterior mean estimates which lie closer to the true parameter values (improving accuracy).

In Appendix D.1, we show experimentally that these results hold for the Heckman special case of our general model. Here we show that our theoretical results apply beyond the Heckman special case by conducting experiments on models with binary outcomes and multiple noise distributions. For all experiments, we use the Bayesian inference package Stan (Carpenter et al., 2017), which uses the Hamiltonian Monte Carlo algorithm (Betancourt, 2017). We report results across 200 trials. For each trial, we generate a new dataset from the data generating process the model assumes; fit the model to that dataset; and evaluate model fit using two metrics: *precision* (width of the 95% confidence interval) and *accuracy* (difference between the posterior mean and the true parameter value). We wish to assess the effect of the constraints on model inferences. Thus, we compare inferences from models with: (i) no constraints (unconstrained); (ii) a prevalence constraint; and (iii) an expertise constraint on a subset of the features. Details are in Appendix D and the code is at https://github.com/sidhikabalachandar/domain_constraints.

Figure 2 shows results for the Bernoulli-sigmoid model with uniform unobservables (equation 4). Both constraints generally produce more precise and accurate inferences for all parameters relative to

the unconstrained model. The one exception is that the expertise constraint does not improve accuracy for $\sigma^2$. Overall, the synthetic experiments corroborate and extend the theoretical analysis, showing that the proposed constraints improve precision and accuracy of parameter estimates for several variants of our general model. (In Appendix D, we also provide results for other variants of our general model, including alternate distributions of unobservables (Figures S2 and S3); higher-dimensional features (Figure S4); and non-linear interactions between features (Figure S5).)

## 5 REAL-WORLD CASE STUDY: BREAST CANCER TESTING

To demonstrate our model's applicability to healthcare settings, we apply it to a breast cancer testing dataset. In this setting, $X_i$ consists of features capturing the person's demographics, genetics, and medical history; $T_i \in \{0, 1\}$ denotes whether a person has been tested for breast cancer; and $Y_i \in \{0, 1\}$ denotes whether the person is diagnosed with breast cancer. Our goal is to learn each person's risk of cancer—i.e., $p(Y_i = 1|X_i)$. We focus on a younger population (age $\leq 45$) because it creates a challenging distribution shift between the tested and untested populations. Younger people are generally not tested for cancer (Cancer Research UK, 2023), so the tested population ($T_i = 1$) may differ from the untested population, including on unobservables.

In the following sections, we describe our experimental set up and the model we fit (§5.1), we conduct four validations on the fitted model (§5.2), we use the model to assess historical testing decisions (§5.3), and we compare to a model fit without a prevalence constraint (§5.4).

### 5.1 EXPERIMENTAL SETUP

Our data comes from the UK Biobank (Sudlow et al., 2015), which contains information on health, demographics, and genetics for the UK (see Appendix E for details). We analyze 54,746 people by filtering for women under the age of 45 (there is no data on breast cancer tests for men). For each person, $X_i$ consists of 7 health, demographic, and genetic features found to be predictive of breast cancer (NIH National Cancer Institute, 2017; Komen, 2023; Yanes et al., 2020). $T_i \in \{0, 1\}$ denotes whether the person receives a mammogram (the most common breast cancer test) in the 10 years following measurement of features. $Y_i \in \{0, 1\}$ denotes whether the person is diagnosed with breast cancer in the 10 year period. $p(T = 1) = 0.51$ and $p(Y = 1|T = 1) = 0.03$.[1]

As in the synthetic experiments, we fit the Bernoulli-sigmoid model with uniform unobservables (equation 4). We include a prevalence constraint $\mathbb{E}[Y] = 0.02$, based on previously reported breast cancer incidence statistics (Cancer Research UK). We also include an expertise constraint by allowing $\beta_{\Delta}$ to deviate from 0 only for features which plausibly influence a person's probability of being tested beyond disease risk. We do not place the expertise constraint on (i) racial/socioeconomic features, due to disparities in healthcare access (Chen et al., 2021; Pierson, 2020; Shanmugam & Pierson, 2021); (ii) genetic features, since genetic information may be unknown or underused (Samphao et al., 2009); and (iii) age, due to age-based breast cancer testing policies (Cancer Research UK, 2023). In Appendix F.2 Figures S7, S8, and S9, we run robustness experiments.

In Figure 3, we plot the inferred coefficients for the fitted model. The model infers a large $\sigma^2 = 5.1$ (95% CI, 3.7-6.8), highlighting the importance of unobservables. In Appendix F.1 Figure S6, we also compare our model's performance to a suite of additional baselines, including (i) baselines trained solely on the tested population, (ii) baselines which treat the untested population as negative, and (iii) additional baselines commonly used in selective labels settings (Rastogi et al., 2023). Collectively, these baselines all suffer from various issues our model does not, including learning implausible age trends inconsistent with prior literature or worse predictive performance.

### 5.2 VALIDATING THE MODEL

Validating models in real-world selective labels settings is difficult because outcomes are not observed for the untested. Still, we leverage the rich data in the UK Biobank to validate our model in four ways.

---

[1]We verify that very few people in the dataset have $T = 0$ and $Y = 1$ (i.e., are diagnosed with no record of a test): $p(Y = 1|T = 0) = 0.0005$. We group these people with the untested $T = 0$ population, since they did not receive a breast cancer test.

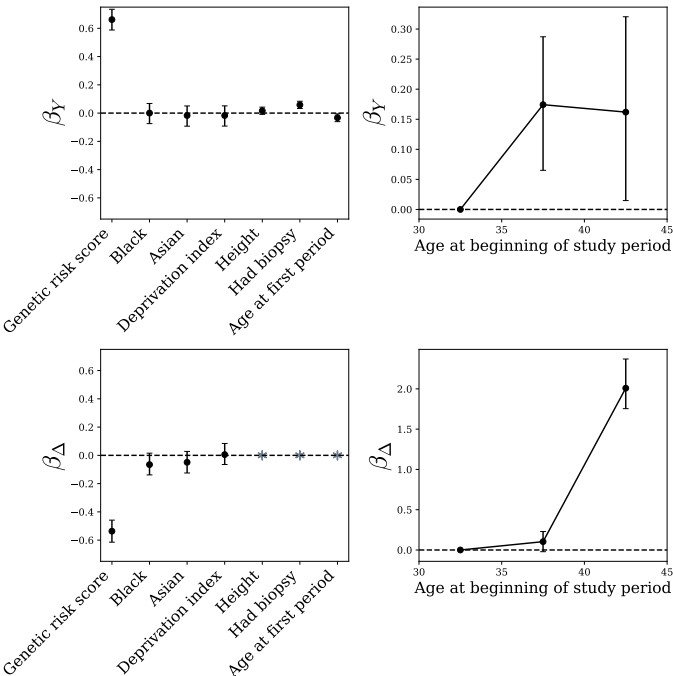

Figure 3: Estimated $\beta_Y$ (top) capture known cancer risk factors: genetic risk, previous biopsy, age at first period (menarche), and age (NIH National Cancer Institute, 2017; Yanes et al., 2020). Estimated $\beta_\Delta$ (bottom) capture the underuse of genetic information (left) and known age-based testing policies (right). Points indicate posterior means and vertical lines indicate 95% confidence intervals. Gray asterisks indicate coefficients set to 0 by the expertise constraint.

**Inferred risk predicts breast cancer diagnoses:** Verifying that inferred risk predicts diagnoses among the *tested* population is straightforward. Since $Y$ is observed for the tested population, we check (on a test set) whether people with higher inferred risk ($p(Y_i = 1|X_i)$) are more likely to be diagnosed with cancer ($Y_i = 1$). People in the highest inferred risk quintile[2] have $3.3\times$ higher true risk of cancer than people in the lowest quintile (6.0% vs 1.8%). Verifying that inferred risk predicts diagnoses among the *untested* population is less straightforward because $Y_i$ is not observed. We leverage that a subset have a *follow-up* visit (i.e., an observation after the initial 10-year study period) to show that inferred risk predicts cancer diagnosis at the follow-up. For the subset of the untested population who attend a follow-up visit, people in the highest inferred risk quintile have $2.5\times$ higher true risk of cancer during the follow-up period than people in the lowest quintile (4.1% vs 1.6%).[3]

**Inferred unobservables correlate with known unobservables:** For each person, our model infers a posterior over unobservables $p(Z_i|X_i, T_i, Y_i)$. We confirm that the inferred posterior mean of unobservables correlates with a true unobservable—whether the person has a family history of breast cancer. This is an unobservable because it influences both $T_i$ and $Y_i$ but is not included in the data given to the model.[4] People in the highest inferred unobservables quintile are $2.1\times$ likelier to have a family history of cancer than people in the lowest quintile (15.6% vs 7.5%).

**$\beta_Y$ captures known cancer risk factors:** $\beta_Y$ measures each feature's contribution to risk. The top left plot in Figure 3 shows that the inferred $\beta_Y$ captures known cancer risk factors. Cancer risk is strongly correlated with genetic risk, and is also correlated with previous breast biopsy, age, and younger age at first period (menarche) (NIH National Cancer Institute, 2017; Yanes et al., 2020).

---

[2]Reporting outcome rates by inferred risk quintile or decile is a common metric in health risk prediction settings (Mullainathan & Obermeyer, 2022; Einav et al., 2018; Obermeyer et al., 2019).

[3]AUC amongst the tested population is 0.63 and amongst the untested population that attended a followup is 0.63. These AUCs are similar to past predictions which use similar feature sets (Yala et al., 2021). For instance, the Tyrer-Cuzick (Tyrer et al., 2004) and Gail (Gail et al., 1989) models achieved AUCs of 0.62 and 0.59.

[4]Although UKBB has family history data, we do not include it as a feature both so we can use it as validation and because we do not have information on *when* family members are diagnosed. So we cannot be sure that the measurement of family history precedes the measurement of $T_i$ and $Y_i$, as is desirable for features in $X_i$.

$\boldsymbol{\beta_\Delta}$ **captures known public health policies:** In the UK, all women aged 50-70 are invited for breast cancer testing every 3 years (Cancer Research UK, 2023). Our study period spans 10 years, so we expect women who are 40 or older at the start of the study period (50 or older at the end) to have an increased probability of testing when controlling for true cancer risk. The bottom right plot in Figure 3 shows this is the case, since the $\boldsymbol{\beta_\Delta}$ indicator for ages 40-45 is greater than the indicators for ages <35 and 35-39.

## 5.3 ASSESSING HISTORICAL TESTING DECISIONS

Non-zero components of $\boldsymbol{\beta_\Delta}$ indicate features that affect a person's probability of being tested even when controlling for their disease risk. The bottom left plot in Figure 3 plots the inferred $\boldsymbol{\beta_\Delta}$, revealing that genetic information is underused. While genetic risk is strongly predictive of $Y_i$, its negative $\boldsymbol{\beta_\Delta}$ indicates that people at high genetic risk are tested less than expected given their risk. This is plausible, given that their genetic information may not have been available to guide decision-making. The model also infers negative point estimates for $\boldsymbol{\beta_\Delta}$ for Black and Asian women, consistent with known racial disparities in breast cancer testing (Makurumidze et al., 2022) as well as broader racial inequality in healthcare and other domains (Nazroo et al., 2007; Zink et al., 2023; Movva et al., 2023; Obermeyer et al., 2019; Franchi et al., 2023; Otu et al., 2020; Devonport et al., 2023). However, both confidence intervals overlap zero (due to the small size of these groups in our dataset).

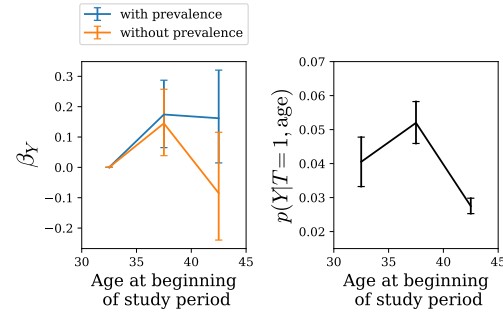

Figure 4: Without the prevalence constraint, the model learns that cancer risk first increases and then decreases with age (left orange), contradicting prior literature (Komen, 2023; Cancer Research UK; US Cancer Statistics Working Group et al., 2013; Campisi, 2013). This incorrect inference occurs because the tested population has the same misleading age trend (right). In contrast, the prevalence constraint encodes that the (younger) untested population has lower risk, allowing the model to learn a more accurate age trend (left blue).

## 5.4 COMPARISON TO MODEL WITHOUT PREVALENCE CONSTRAINT

The prevalence constraint also guides the model to more plausible inferences. We compare the model fit with and without a prevalence constraint. As shown in the left plot in Figure 4, without the prevalence constraint, the model learns that cancer risk first increases with age and then falls, contradicting prior epidemiological and physiological evidence (Komen, 2023; Cancer Research UK; US Cancer Statistics Working Group et al., 2013; Campisi, 2013). This is because, due to the age-based testing policy in the UK (Cancer Research UK, 2023), being tested for breast cancer before age 50 is unusual. Thus, the tested population under age 50 is non-representative because their risk is much higher than the corresponding untested population. The prevalence constraint guides the model to more plausible inferences by preventing the model from predicting that a large fraction of the untested (younger) population has the disease.

## 6 RELATED WORK

Selective labels problems occur in many domains, including hiring, insurance, government inspections, tax auditing, recommender systems, lending, healthcare, education, welfare services, wildlife protection, and criminal justice (Lakkaraju et al., 2017; Jung et al., 2020a; Kleinberg et al., 2018; Björkegren & Grissen, 2020; Jung et al., 2018; Jehi et al., 2020; McDonald et al., 2021; Laufer et al.; McWilliams et al., 2019; Crook & Banasik, 2004; Hong et al., 2018; Parker et al., 2019; Sun et al., 2011; Kansagara et al., 2011; Waters & Miikkulainen, 2014; Bogen, 2019; Jawaheer et al., 2010; Wu et al., 2017; Coston et al., 2020; De-Arteaga et al., 2021; Pierson, 2020; Pierson et al., 2020; Simoiu et al., 2017; Mullainathan & Obermeyer, 2022; Henderson et al., 2022; Gholami et al., 2019; Farahani et al., 2020; Liu & Garg, 2022; Cai et al., 2020; Daysal et al., 2022; Guerdan et al., 2023; Chan et al., 2022; Jiang et al., 2021; Chien et al., 2023; Jia et al., 2019). As such, there are related literatures

in machine learning and causal inference (Coston et al., 2020; Schulam & Saria, 2017; Lakkaraju et al., 2017; Kleinberg et al., 2018; Shimodaira, 2000; De-Arteaga et al., 2021; Levine et al., 2020; Koh et al., 2021; Sagawa et al., 2022; Kaur et al., 2022; Sahoo et al., 2022; Cortes-Gomez et al., 2023), econometrics (Mullainathan & Obermeyer, 2022; Rambachan et al., 2022; Heckman, 1976; Hull, 2021; Künzel et al., 2019; Shalit et al., 2017; Wager & Athey, 2018; Alaa & Schaar, 2018), statistics and Bayesian models (Ilyas et al., 2020; Daskalakis et al., 2021; Mishler & Kennedy, 2022; Jung et al., 2020b), and epidemiology (Groenwold et al., 2012; Perkins et al., 2018). We extend this literature by providing constraints which both theoretically and empirically improve parameter inference. We now describe the three lines of work most closely related to our modeling approach.

**Generalized linear mixed models (GLMMs):**  Our model is closely related to GLMMs (Gelman et al., 2013; Stroup, 2012; Lum et al., 2022), which model observations as a function of both observed features $X_i$ and unobserved "random effects" $Z_i$. We extend this literature by (i) proposing and analyzing a novel model to capture our selective labels setting; (ii) incorporating the uniform distribution of unobservables, as opposed to the normal distribution typically used in GLMMs, to yield more tractable inference; and most importantly (iii) incorporating healthcare domain constraints into GLMMs to improve model estimation.

**Improving robustness to distribution shift using domain information:**  The selective labels setting represents a specific type of distribution shift from the tested to untested population. Previous work shows that generic methods often fail to perform well across all types of distribution shifts (Gulrajani & Lopez-Paz, 2021; Koh et al., 2021; Sagawa et al., 2022; Wiles et al., 2022; Kaur et al., 2022) and that incorporating domain information can improve performance. Gao et al. (2023) proposes *targeted augmentations*, which augment the data by randomizing known spurious features while preserving robust ones. Tellez et al. (2019) presents an example of this strategy for histopathology slide analysis. Kaur et al. (2022) shows that modeling the data generating process is necessary for generalizing across distribution shifts. Motivated by this, we propose a data generating process suitable for selective labels settings and show that using domain information improves performance.

**Breast cancer risk estimation:**  There are many related works on estimating breast cancer risk (Daysal et al., 2022; Yala et al., 2019; 2021; 2022; Shen et al., 2021). Our work complements this literature by proposing a Bayesian model which captures the selective labels setting and incorporating domain constraints to improve model estimation. While a linear model suffices for the low-dimensional features used in our case study, our approach naturally extends to more complex inputs (e.g., medical images) and deep learning models sometimes used in breast cancer risk prediction (Yala et al., 2019; 2021; 2022).

## 7  DISCUSSION

We propose a Bayesian model class to infer risk and assess historical human decision-making in selective labels settings, which commonly occur in healthcare and other domains. We propose the prevalence and expertise constraints which we show both theoretically and empirically improve parameter inference. We apply our model to cancer risk prediction, validate its inferences, show it can identify suboptimalities in test allocation, and show the prevalence constraint prevents misleading inferences.

A natural future direction is applying our model to other healthcare settings, where a frequent practice is to train risk-prediction models only on the tested population (Jehi et al., 2020; McDonald et al., 2021; Farahani et al., 2020). This is far from optimal both because only a small fraction of the population is tested, increasing variance, and because the tested population is highly non-representative, increasing bias. The paradigm we propose offers a solution to both problems. Using data from the entire population reduces variance, and modeling the distribution shift and constraining inferences on the untested population reduces bias. Beyond healthcare, other selective labels domains may have other natural domain constraints: for example, randomly assigned human decision-makers (Kleinberg et al., 2018) or repeated measurements of the same individual (Lum et al., 2022). Beyond selective labels, our model represents a concrete example of how domain constraints can improve inference in the presence of distribution shift.

ACKNOWLEDGMENTS

The authors thank Gabriel Agostini, Sivaramakrishnan Balachandar, Serina Chang, Erica Chiang, Avi Feller, Eran Halperin, Andrew Ilyas, Pang Wei Koh, Ben Laufer, Zhi Liu, Smitha Milli, Sendhil Mullainathan, Josue Nassar, Kenny Peng, Ashesh Rambachan, Richa Rastogi, Evan Rose, Shuvom Sadhuka, Jacob Steinhardt, Robert Tillman, and Manolis Zampetakis for helpful conversations. This research was supported by a Google Research Scholar award, NSF CAREER #2142419, a CIFAR Azrieli Global scholarship, Optum, a LinkedIn Research Award, the Abby Joseph Cohen Faculty Fund, and NSF GRFP Grant DGE #2139899. This research has been conducted using the UK Biobank Resource under Application Number 72589. Any opinions, findings, conclusions, or recommendations expressed in this material are those of the authors and do not necessarily reflect the views of the funders.

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
