## A    CALCULATING DISEASE PREVALENCE

To implement the prevalence constraint, we assume that the *disease prevalence*, or average value of $Y$ across the population, is at least approximately known. This assumption is plausible in medical settings because estimating prevalence is the focus of substantial public health research. Methods to calculate prevalence include serology, where blood samples are used to detect specific antibodies or antigens of a disease (Joseph et al., 1995); stool or wastewater testing for disease markers (Joseph et al., 1995; McMahan et al., 2021); genetic methods, where genomic registries can be analyzed to calculate allele frequency and estimate disease prevalence (Schrodi et al., 2015); autopsy reports for a particular disease (Bell et al., 2015); and administrative data collected by primary, outpatient, and inpatient care centers (Wiréhn et al., 2007). Additionally, our Bayesian formulation can incorporate approximate prevalence estimates (e.g. bounded estimates), and these bounds can be estimated using the sensitivity and specificity of the prevalence estimation method (Manski & Molinari, 2021; Manski, 2020; Mullahy et al., 2021).

## B    PROOFS

**Proof outline:**   In this section, we provide three proofs to show why domain constraints improve parameter inference. We start by showing that the well-studied Heckman correction model (Heckman, 1976; 1979) is a special case of the general model in equation 1 (Proposition 3.1). It is known that placing constraints on the Heckman model can improve parameter inference (Lewbel, 2019). We show that our proposed prevalence and expertise constraints have a similar effect by proving that our proposed constraints never worsen the precision of parameter inference (Proposition 3.2). We then provide conditions under which our constraints strictly improve precision (Proposition B.2).

**Notation and assumptions:**   Below, we use $\Phi$ to denote the normal CDF, $\phi$ the normal PDF, and $\boldsymbol{\beta_T} = \alpha\boldsymbol{\beta_Y} + \boldsymbol{\beta_\Delta}$. Let $X$ be the matrix of observable features. We assume that the first column of $X$ corresponds to the intercept; $X$ is zero mean for all columns except the intercept; and the standard identifiability condition that our data matrix is full rank, i.e., $X^T X$ is invertible. We also assume that $\alpha > 0$.

We start by defining the Heckman correction model.

**Definition 1** (Heckman correction model). *The Heckman model can be written in the following form (Hicks, 2021):*

$$T_i = \mathbb{1}[X_i^T \tilde{\boldsymbol{\beta}}_{\boldsymbol{T}} + u_i > 0]$$
$$Y_i = X_i^T \tilde{\boldsymbol{\beta}}_{\boldsymbol{Y}} + Z_i$$
$$\begin{bmatrix} u_i \\ Z_i \end{bmatrix} \sim Normal\left( \begin{bmatrix} 0 \\ 0 \end{bmatrix}, \begin{bmatrix} 1 & \tilde{\rho} \\ \tilde{\rho} & \tilde{\sigma}^2 \end{bmatrix} \right). \tag{2}$$

In other words, $T_i = 1$ if a linear function of $X_i$ plus some unit normal noise $u_i$ exceeds zero. $Y_i$ is a linear function of $X_i$ plus normal noise $Z_i$ with variance $\tilde{\sigma}^2$. Importantly, the noise terms $Z_i$ and $u_i$ are *correlated*, with covariance $\tilde{\rho}$. The model parameters are $\tilde{\theta} \triangleq (\tilde{\rho}, \tilde{\sigma}^2, \tilde{\boldsymbol{\beta}}_{\boldsymbol{T}}, \tilde{\boldsymbol{\beta}}_{\boldsymbol{Y}})$. We use tildes over the Heckman model parameters to distinguish them from the parameters in our original model in equation 1. We now prove Proposition 3.1.

**Proposition 3.1.** *The Heckman model (Definition 1) is equivalent to the following special case of the general model in equation 1:*

$$Z_i \sim \mathcal{N}(0, \sigma^2)$$
$$r_i = X_i^T \boldsymbol{\beta_Y} + Z_i$$
$$Y_i = r_i \tag{3}$$
$$T_i \sim Bernoulli(\Phi(\alpha r_i + X_i^T \boldsymbol{\beta_\Delta})).$$

*Proof.* If we substitute in the value of $r_i$, the equation for $Y_i$ is equivalent to that in the Heckman model. So it remains only to show that $T_i$ in equation 3 can be rewritten in the form in equation 2.

We first rewrite equation 3 in slightly more convenient form:

$$T_i \sim \text{Bernoulli}(\Phi(\alpha r_i + X_i^T \boldsymbol{\beta_\Delta})) \to$$
$$T_i \sim \text{Bernoulli}(\Phi(\alpha(X_i^T \boldsymbol{\beta_Y} + Z_i) + X_i^T \boldsymbol{\beta_\Delta})) \to$$
$$T_i \sim \text{Bernoulli}(\Phi(X_i^T(\alpha \boldsymbol{\beta_Y} + \boldsymbol{\beta_\Delta}) + \alpha Z_i)) \to$$
$$T_i \sim \text{Bernoulli}(\Phi(X_i^T \boldsymbol{\beta_T} + \alpha Z_i)).$$

We then apply the latent variable formulation of the probit link:

$$T_i \sim \text{Bernoulli}(\Phi(X_i^T \boldsymbol{\beta_T} + \alpha Z_i)) \to$$
$$T_i = \mathbb{1}[X_i^T \boldsymbol{\beta_T} + \alpha Z_i + \epsilon_i > 0], \epsilon_i \sim \mathcal{N}(0,1),$$

where $\alpha Z_i + \epsilon_i$ is a normal random variable with standard deviation $\sqrt{\alpha^2 \sigma^2 + 1}$. We divide through by this factor to rewrite the equation for $T_i$:

$$T_i = \mathbb{1}[X_i^T \tilde{\boldsymbol{\beta}}_T + u_i > 0],$$

which is equivalent to equation 2. Here, $\tilde{\boldsymbol{\beta}}_T = \frac{\beta_T}{\sqrt{\alpha^2 \sigma^2 + 1}}$ and $u_i = \frac{\alpha Z_i + \epsilon_i}{\sqrt{\alpha^2 \sigma^2 + 1}}$ is a unit-scale normal random variable whose covariance with $Z_i$ is

$$\text{cov}\left(\frac{\alpha Z_i + \epsilon_i}{\sqrt{\alpha^2 \sigma^2 + 1}}, Z_i\right) = \mathbb{E}\left(\frac{\alpha Z_i + \epsilon_i}{\sqrt{\alpha^2 \sigma^2 + 1}} \cdot Z_i\right) - \mathbb{E}\left(\frac{\alpha Z_i + \epsilon_i}{\sqrt{\alpha^2 \sigma^2 + 1}}\right) \mathbb{E}(Z_i)$$
$$= \frac{\alpha \mathbb{E}(Z_i^2)}{\sqrt{\alpha^2 \sigma^2 + 1}}$$
$$= \frac{\alpha \sigma^2}{\sqrt{\alpha^2 \sigma^2 + 1}}.$$

Thus, the special case of our model in equation 3 is equivalent to the Heckman model, where the mapping between the parameters is:

$$\tilde{\boldsymbol{\beta}}_Y = \boldsymbol{\beta_Y}$$
$$\tilde{\sigma}^2 = \sigma^2$$
$$\tilde{\boldsymbol{\beta}}_T = \frac{\boldsymbol{\beta_T}}{\sqrt{\alpha^2 \sigma^2 + 1}} \qquad (5)$$
$$\tilde{\rho} = \frac{\alpha \sigma^2}{\sqrt{\alpha^2 \sigma^2 + 1}}.$$

$\square$

As described in Lewbel (2019), the Heckman correction model is identified without any further assumptions. It then follows that the special case of our model in equation 3 is identified without further constraints. One can simply estimate the Heckman model, which by the mapping in equation 5 immediately yields estimates of $\boldsymbol{\beta_Y}$ and $\sigma^2$. Then, the equation for $\tilde{\rho}$ can be solved for $\alpha$, yielding a unique value since $\alpha > 0$. Similarly the equation for $\tilde{\boldsymbol{\beta}}_T$ yields the estimate for $\boldsymbol{\beta_T}$ (and thus $\boldsymbol{\beta_\Delta}$).

While the Heckman model is identified without further constraints, this identification is known to be very weak, relying on functional form assumptions (Lewbel, 2019). To mitigate this problem, when the Heckman model is used in the econometrics literature it is typically estimated with constraints on the parameters. In particular, a frequently used constraint is an *exclusion restriction*: there must be at least one feature with a non-zero coefficient in the equation for $T$ but not $Y$. While this constraint differs from the ones we propose, one might expect our proposed prevalence and expertise constraints to have a similar effect and improve the precision of parameter inference. We make this precise through Proposition 3.2.

Throughout the results below, we analyze the posterior distribution of model parameters given the observed data: $g(\theta) \triangleq p(\theta|X, T, Y)$. We show that constraining the value of any one parameter (through the prevalence or expertise constraint) will not worsen the posterior variance of the other parameters. In particular, constraining a parameter $\theta_{\text{con}}$ to a value drawn from its posterior distribution will not in expectation increase the posterior variance of any other unconstrained parameters $\theta_{\text{unc}}$. To formalize this, we define the *expected conditional variance*:

**Definition 2** (Expected conditional variance). *Let the distribution over model parameters $g(\theta) \triangleq p(\theta|X, T, Y)$ be the posterior distribution of the parameters $\theta$ given the observed data $\{X, T, Y\}$. We define the expected conditional variance of an unconstrained parameter $\theta_{unc}$, conditioned on the value of a constrained parameter $\theta_{con}$, to be $\mathbb{E}[Var(\theta_{unc}|\theta_{con})] \triangleq \mathbb{E}_{\theta_{con}^* \sim g}[Var(\theta_{unc}|\theta_{con} = \theta_{con}^*)]$.*

**Proposition 3.2.** *In expectation, constraining the parameter $\theta_{con}$ does not increase the variance of any other parameter $\theta_{unc}$. In other words, $\mathbb{E}[Var(\theta_{unc}|\theta_{con})] \leq Var(\theta_{unc})$. Moreover, the inequality is strict as long as $\mathbb{E}[\theta_{unc}|\theta_{con}]$ is non-constant in $\theta_{con}$ (i.e., $Var(\mathbb{E}[\theta_{unc}|\theta_{con}]) > 0$).*

*Proof.* The proof follows from applying the law of total variance to the posterior distribution $g$. The law of total variance states that:

$$\text{Var}(\theta_{\text{unc}}) = \mathbb{E}[\text{Var}(\theta_{\text{unc}}|\theta_{\text{con}})] + \text{Var}(\mathbb{E}[\theta_{\text{unc}}|\theta_{\text{con}}]) \,.$$

Since $\text{Var}(\mathbb{E}[\theta_{\text{unc}}|\theta_{\text{con}}])$ is non-negative,

$$\mathbb{E}[\text{Var}(\theta_{\text{unc}}|\theta_{\text{con}})] \leq \text{Var}(\theta_{\text{unc}}) \,.$$

Additionally, if $\mathbb{E}[\theta_{\text{unc}}|\theta_{\text{con}}]$ is non-constant in $\theta_{\text{con}}$ then $\text{Var}(\mathbb{E}[\theta_{\text{unc}}|\theta_{\text{con}}])$ is strictly positive. Thus the strict inequality follows. □

We now discuss how Proposition 3.2 applies to our proposed constraints and the Heckman model. Both the prevalence and expertise constraints fix the value of at least one parameter. The prevalence constraint fixes the value of $\boldsymbol{\beta_{Y}}_0$ and the expertise constraint fixes the value of $\boldsymbol{\beta_{\Delta}}_d$ for some $d$. Thus by Proposition 3.2, we know that the prevalence and expertise constraints will not increase the variance of any model parameters, and will strictly reduce them as long as the posterior expectations of the unconstrained parameters are non-constant in the constrained parameters.

We now show that when $\tilde{\boldsymbol{\beta}}_T$ is known, the prevalence constraint strictly reduces variance. The setting where $\tilde{\boldsymbol{\beta}}_T$ is known is a natural one because $\tilde{\boldsymbol{\beta}}_T$ can be immediately estimated from the observed data $X$ and $T$, and previous work in both econometrics and statistics thus have also considered this setting (Heckman, 1976; Ilyas et al., 2020). With additional assumptions, we also show that the expertise constraint strictly reduces variance. We derive these results in the setting with flat priors for algebraic simplicity. However, analogous results also hold under other natural choices of prior (e.g., standard conjugate priors for Bayesian linear regression (Jackman, 2009)). In the results below, we analyze the conditional mean of $Y$ conditioned on $T = 1$. Thus, we start by defining this value.

**Lemma B.1** (Conditional mean of $Y$ conditioned on $T = 1$). *Past work has shown that the expected value of $Y_i$ when $T_i = 1$ is (Hicks, 2021):*

$$\mathbb{E}[Y_i|T_i = 1] = \mathbb{E}[Y_i|X_i^T \tilde{\boldsymbol{\beta}}_T + u > 0]$$

$$= X_i \tilde{\boldsymbol{\beta}}_Y + \tilde{\rho}\tilde{\sigma} \frac{\phi(X_i \tilde{\boldsymbol{\beta}}_T)}{\Phi(X_i \tilde{\boldsymbol{\beta}}_T)} \,,$$

*where $\Phi$ denotes the normal CDF, $\phi$ the normal PDF, and $\frac{\phi(X \tilde{\boldsymbol{\beta}}_T)}{\Phi(X \tilde{\boldsymbol{\beta}}_T)}$ the inverse Mills ratio. This can be more succinctly represented in matrix notation as*

$$\mathbb{E}[Y_i|T_i = 1] = M\theta \,,$$

*where $M = [X_{T=1}; \frac{\phi(X_{T=1} \tilde{\boldsymbol{\beta}}_T)}{\Phi(X_{T=1} \tilde{\boldsymbol{\beta}}_T)}] \in \mathbb{R}^{N_{T=1} \times (d+1)}$, $\theta = [\tilde{\boldsymbol{\beta}}_Y, \tilde{\rho}\tilde{\sigma}] \in \mathbb{R}^{d+1}$, $X_{T=1}$ denotes the rows of $X$ corresponding to $T = 1$, and $N_{T=1}$ is the number of rows of $X$ for which $T = 1$.*

**Proposition B.2.** *Assume $\tilde{\boldsymbol{\beta}}_T$ is fixed and flat priors on all parameters. Additionally, assume the standard identifiability condition that the matrix $M = [X_{T=1}; \frac{\phi(X_{T=1} \tilde{\boldsymbol{\beta}}_T)}{\Phi(X_{T=1} \tilde{\boldsymbol{\beta}}_T)}]$ is full rank. Then, in expectation, constraining a component of $\tilde{\boldsymbol{\beta}}_Y$ in the Heckman correction model strictly reduces the posterior variance of the other model parameters. The prevalence constraint does this without any further assumptions, and the expertise constraint does this if $\tilde{\rho}$ and $\tilde{\sigma}^2$ are fixed.*

*Proof.* We will start by showing that when $\tilde{\boldsymbol{\beta}}_T$ is fixed, constraining a component of $\tilde{\boldsymbol{\beta}}_Y$ strictly reduces the variance of the other model parameters. From the definition of the conditional mean of $Y$ conditioned on $T = 1$ (Lemma B.1), we get

$$\mathbb{E}[Y_i|T_i = 1] = M\theta \,.$$

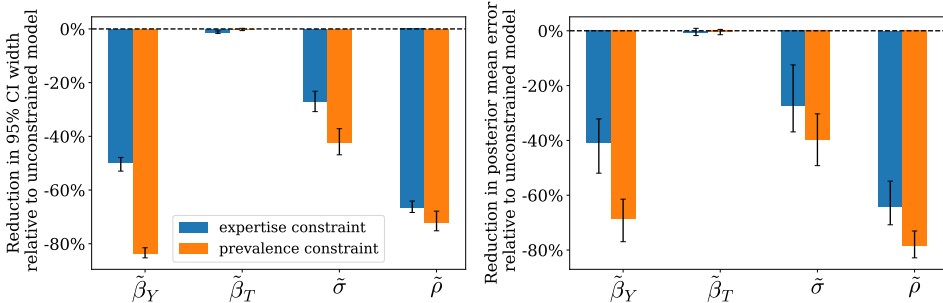

Figure S1: Results using synthetic data from the Heckman model. The prevalence and expertise constraints each produce more precise and accurate inferences on this synthetic data. We plot the median across 200 synthetic datasets. Errorbars denote the bootstrapped 95% confidence interval on the median.

Under flat priors on all parameters, the posterior expectation of the model parameters given the observed data $\{X, T, Y\}$ is simply the standard ordinary least squares solution given by the normal equation (Jackman, 2009):

$$\mathbb{E}[\theta|X, T, Y] = (M^T M)^{-1} M^T Y .$$

By assumption, $M$ is full rank, so $M^T M$ is invertible.

When $\tilde{\boldsymbol{\beta}}_{\boldsymbol{Y}_d}$ is constrained to equal to $\tilde{\boldsymbol{\beta}}^*_{\boldsymbol{Y}_d}$ for some component $d$, the equation instead becomes:

$$\mathbb{E}[\theta_{-d}|\tilde{\boldsymbol{\beta}}_{\boldsymbol{Y}_d} = \tilde{\boldsymbol{\beta}}^*_{\boldsymbol{Y}_d}, X, T, Y] = (M^T_{-d} M_{-d})^{-1} M^T_{-d}(Y - X_{T=1_d}\tilde{\boldsymbol{\beta}}^*_{\boldsymbol{Y}_d}) .$$

We use the subscript $-d$ notation to indicate that we no longer estimate the component $d$. Here, $M_{-d} = [X_{T=1_{-d}}; \frac{\phi(X_{T=1}\tilde{\boldsymbol{\beta}}_{\boldsymbol{T}})}{\Phi(X_{T=1}\tilde{\boldsymbol{\beta}}_{\boldsymbol{T}})}] \in \mathbb{R}^{N_{T=1} \times d}$ and $\theta_{-d} = [\tilde{\boldsymbol{\beta}}_{\boldsymbol{Y}_{-d}}, \tilde{\rho}\tilde{\sigma}] \in \mathbb{R}^d$. Since $X_{T=1_d}$ is nonzero and $M$ is full rank, it follows that $\mathbb{E}[\theta_{-d}|\tilde{\boldsymbol{\beta}}_{\boldsymbol{Y}_d} = \tilde{\boldsymbol{\beta}}^*_{\boldsymbol{Y}_d}, X, T, Y]$ is not constant in $\tilde{\boldsymbol{\beta}}^*_{\boldsymbol{Y}_d}$. Thus by Proposition 3.2, constraining $\tilde{\boldsymbol{\beta}}_{\boldsymbol{Y}_d}$ reduces the variance of the parameters in $\theta_{-d}$ ($\tilde{\boldsymbol{\beta}}_{\boldsymbol{Y}_{d'}}$ for $d' \neq d$ and $\tilde{\rho}\tilde{\sigma}$).

We will now show that both the prevalence and expertise constraints constrain a component of $\tilde{\boldsymbol{\beta}}_{\boldsymbol{Y}}$. Assuming the standard condition that columns of $X$ are zero-mean except for an intercept column of ones, the prevalance constraint fixes

$$\mathbb{E}_Y[Y] = \mathbb{E}_Y[\mathbb{E}_X[\mathbb{E}_Z[Y|X, Z]]]$$
$$= \mathbb{E}_X[\mathbb{E}_Z[X^T\boldsymbol{\beta_Y} + Z]]$$
$$= \boldsymbol{\beta_Y}_0 ,$$

where $\boldsymbol{\beta_Y}_0$ is the 0th index (intercept term) of $\boldsymbol{\beta_Y}$. The expertise constraint also fixes a component of $\tilde{\boldsymbol{\beta}}_{\boldsymbol{Y}}$ if $\tilde{\rho}$ and $\tilde{\sigma}^2$ are fixed. This can be shown by algebraically rearranging equation 5 to yield

$$\tilde{\boldsymbol{\beta}}_{\boldsymbol{Y}} = \tilde{\boldsymbol{\beta}}_{\boldsymbol{T}}\frac{\tilde{\sigma}^2}{\tilde{\rho}} - \boldsymbol{\beta_\Delta}\frac{\tilde{\sigma}\sqrt{\tilde{\sigma}^2 - \tilde{\rho}^2}}{\tilde{\rho}} .$$

$\square$

While we derive our theoretical results for the Heckman correction model, in both our synthetic experiments (§4) and our real-world case study (§5) we validate that our constraints improve parameter inference beyond the special Heckman case.

## C  DERIVATION OF THE CLOSED-FORM UNIFORM UNOBSERVABLES MODEL

Conducting sampling for our general model described by equation 1 is faster if the distribution of unobservables $f$ and link functions $h_Y$ and $h_T$ allow one to marginalize out $Z_i$ through closed-form integrals, since otherwise $Z_i$ must be sampled for each datapoint $i$, producing a high-dimensional

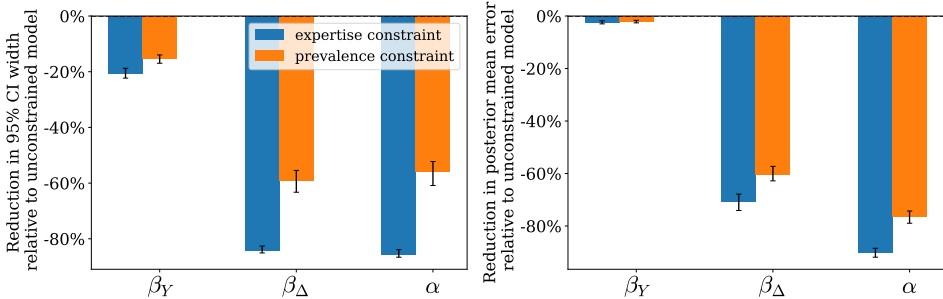

Figure S2: Results using synthetic data from the Bernoulli-sigmoid model with normal unobservables and fixed $\sigma^2$. The prevalence and expertise constraints each produce more precise and accurate inferences on this synthetic data. We plot the median across 200 synthetic datasets. Errorbars denote the bootstrapped 95% confidence interval on the median.

latent variable which slows computation and convergence. Many distributions do not produce closed-form integrals when combined with a sigmoid or probit link function, which are two of the most commonly used links with binary variables.[5] However, we *can* derive closed forms for the special *uniform unobservables* case described by equation 4.

Below, we leave the $i$ subscript implicit to keep the notation concise. When computing the log likelihood of the data, to marginalize out $Z$, we must be able to derive closed forms for the following three integrals:

$$p(Y = 1, T = 1|X) = \int_Z p(Y = 1, T = 1|X, Z)f(Z)dZ$$

$$p(Y = 0, T = 1|X) = \int_Z p(Y = 0, T = 1|X, Z)f(Z)dZ$$

$$p(T = 0|X) = \int_Z p(T = 0|X, Z)f(Z)dZ,$$

since the three possibilities for an individual datapoint are $\{Y = 1, T = 1\}$, $\{Y = 0, T = 1\}$, $\{T = 0\}$. To implement the prevalence constraint (which fixes the $\mathbb{E}[Y]$), we also need a closed form for the following integral:

$$p(Y = 1|X) = \int_Z p(Y = 1|X, Z)f(Z)dZ.$$

For the uniform unobservables model with $\alpha = 1$, the four integrals have the following closed forms, where below we define $A = e^{X^T \beta_T}$ and $B = e^{X^T \beta_Y}$:

$$p(Y = 1, T = 1|X) = \frac{1}{\sigma (A - B)} \bigg( \sigma (A - B) - A \log \left( (B + 1) A^{-1} \right)$$
$$+ A \log \left( (Be^\sigma + 1) A^{-1} e^{-\sigma} \right) + B \log \left( (A + 1) A^{-1} \right)$$
$$- B \log \left( (Ae^\sigma + 1) A^{-1} e^{-\sigma} \right) \bigg)$$

$$p(Y = 0, T = 1|X) = \frac{1}{\sigma (A - B)} \bigg( \left( - \log \left( (A + 1) A^{-1} \right) + \log \left( (B + 1) A^{-1} \right) \right.$$
$$+ \log \left( (Ae^\sigma + 1) A^{-1} e^{-\sigma} \right) - \log \left( (Be^\sigma + 1) A^{-1} e^{-\sigma} \right) \big) A \bigg)$$

$$p(T = 0|X) = \frac{\log \left( 1 + A^{-1} \right) - \log \left( A^{-1} e^{-\sigma} + 1 \right)}{\sigma}$$

$$p(Y = 1|X) = \frac{\sigma - \log \left( 1 + B^{-1} \right) + \log \left( B^{-1} e^{-\sigma} + 1 \right)}{\sigma}.$$

The integrals also have closed forms for other integer values of $\alpha$ (e.g., $\alpha = 2$) allowing one to perform robustness checks with alternate model specifications (see Appendix F.2 Figure S8).

---

[5]Specifically, we search over the distributions in McLaughlin (2001), combined with logit or probit links, and find that most combinations do not yield closed forms.

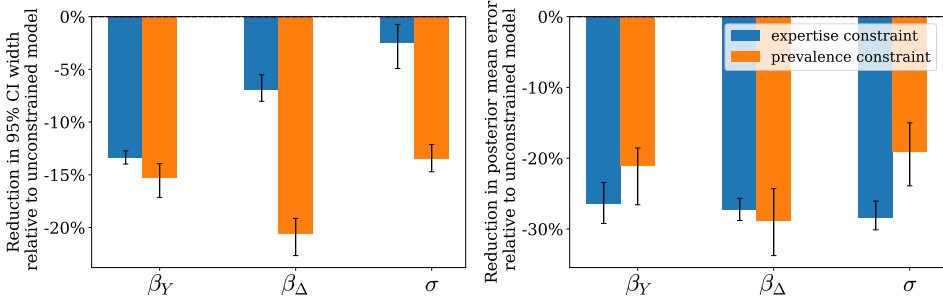

Figure S3: Results using synthetic data from the Bernoulli-sigmoid model with normal unobservables and fixed $\alpha$. The prevalence and expertise constraints each produce more precise and accurate inferences on this synthetic data. We plot the median across 200 synthetic datasets. Errorbars denote the bootstrapped 95% confidence interval on the median.

## D  SYNTHETIC EXPERIMENTS

We first validate that the prevalence and expertise constraints improve the precision and accuracy of parameter inference for the Heckman model described in equation 2. We then extend beyond this special case and examine various Bernoulli-sigmoid instantiations of our general model in equation 1, which assume a binary outcome variable $Y$. With a binary outcome, models are known to be more challenging to fit: for example, one cannot simultaneously estimate both $\alpha$ and $\sigma^2$ (so we must fix either $\alpha$ or $\sigma^2$), and models fit without constraints may fail to recover the correct parameters (StataCorp, 2023; Van de Ven & Van Praag, 1981; Toomet & Henningsen, 2008). We assess whether our proposed constraints improve model estimation even in this more challenging case. Specifically, we extend beyond the Heckman model to the following data generating settings: (i) uniform unobservables and fixed $\alpha$, (ii) normal unobservables and fixed $\sigma^2$; (iii) normal unobservables and fixed $\alpha$; and (iv) other more complex models. For the uniform model, we conduct experiments only with fixed $\alpha$ (not fixed $\sigma^2$) because, as discussed above, this allows us to marginalize out $Z$.

In all models, to incorporate the prevalence constraint into the model, we add a quadratic penalty to the model penalizing it for inferences that produce an inferred $\mathbb{E}[Y]$ that deviates from the true $\mathbb{E}[Y]$. To incorporate the expertise constraint into the model, we set the model parameters $\boldsymbol{\beta}_{\Delta_d}$ to be equal to 0 for all dimensions $d$ to which the expertise constraint applies.

### D.1  HECKMAN MODEL

We first conduct synthetic experiments using the Heckman model defined in equation 2. This model is identifiable without any further constraints, thus we estimate parameters $\theta \triangleq (\tilde{\rho}, \tilde{\sigma}^2, \tilde{\boldsymbol{\beta}}_{\boldsymbol{T}}, \tilde{\boldsymbol{\beta}}_{\boldsymbol{Y}})$.

In the simulation, we use 5000 datapoints; 5 features (including the intercept column of 1s); $X$, $\boldsymbol{\beta}_Y$, and $\boldsymbol{\beta}_T$ drawn from unit normal distributions; and $\sigma \sim \mathcal{N}(2, 0.1)$. We draw the intercept terms $\boldsymbol{\beta}_{Y_0} \sim \mathcal{N}(-2, 0.1)$ and $\boldsymbol{\beta}_{T_0} \sim \mathcal{N}(2, 0.1)$. We assume the expertise constraint applies to $\boldsymbol{\beta}_{\Delta_2} = \boldsymbol{\beta}_{\Delta_3} = \boldsymbol{\beta}_{\Delta_4} = 0$. Thus, by rearranging equation 5, we fix $\tilde{\boldsymbol{\beta}}_{\boldsymbol{Y}} = \tilde{\boldsymbol{\beta}}_{\boldsymbol{T}} \frac{\tilde{\sigma}^2}{\tilde{\rho}}$. When calculating the results for $\tilde{\boldsymbol{\beta}}_{\boldsymbol{T}}$ and $\tilde{\boldsymbol{\beta}}_{\boldsymbol{Y}}$, we do not include the dimensions along which we assume expertise since these dimensions are assumed to be fixed for the model with the expertise constraint.

We show results in Figure S1. Both constraints generally produce more precise and accurate inferences for all parameters relative to the unconstrained model. The only exception is $\tilde{\boldsymbol{\beta}}_{\boldsymbol{T}}$, for which both models produce equivalently accurate and precise inferences. This is consistent with our theoretical results, which do not imply that the precision of inference for $\tilde{\boldsymbol{\beta}}_{\boldsymbol{T}}$ should improve.

### D.2  UNIFORM UNOBSERVABLES MODEL

We now discuss our synthetic experiments using the Bernoulli-sigmoid model with uniform unobservables and $\alpha = 1$ in equation 4. Our simulation parameters are similar to the Heckman model experiments. We use 5000 datapoints; 5 features (including the intercept column of 1s); $X$, $\boldsymbol{\beta}_Y$, and $\boldsymbol{\beta}_\Delta$ drawn from unit normal distributions; and $\sigma \sim \mathcal{N}(2, 0.1)$. We draw the intercept terms $\boldsymbol{\beta}_{Y_0} \sim \mathcal{N}(-2, 0.1)$ and $\boldsymbol{\beta}_{\Delta_0} \sim \mathcal{N}(2, 0.1)$ to approximately match $p(Y)$ and $p(T)$ in realistic medical

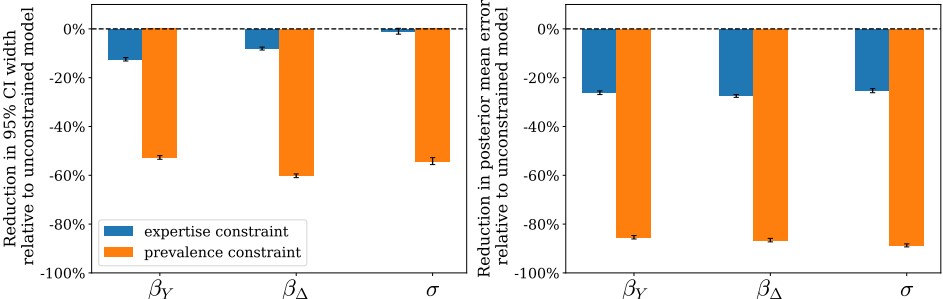

Figure S4: The prevalence and expertise constraints still improve parameter inference when quadrupling the number of features relative to Figure 2. Results are shown using synthetic data from the Bernoulli-sigmoid model with uniform unobservables. Both constraints produce more precise and accurate inferences on this synthetic data. We plot the median across 200 synthetic datasets. Errorbars denote the bootstrapped 95% confidence interval on the median.

settings, where disease prevalence is relatively low, but a large fraction of the population is tested because false negatives are more costly than false positives. We assume the expertise constraint applies to $\boldsymbol{\beta}_{\Delta_2} = \boldsymbol{\beta}_{\Delta_3} = \boldsymbol{\beta}_{\Delta_4} = 0$. We show results in Figure 2. When calculating the results for $\boldsymbol{\beta_\Delta}$, we do not include the dimensions along which we assume expertise since these dimensions are assumed to be fixed for the model with the expertise constraint.

### D.3 NORMAL UNOBSERVABLES MODEL

We also conduct synthetic experiments using the following Bernoulli-sigmoid model with normal unobservables:

$$
\begin{aligned}
Z_i &\sim \mathcal{N}(0, \sigma^2) \\
r_i &= X_i^T \boldsymbol{\beta_Y} + Z_i \\
Y_i &\sim \text{Bernoulli}(\text{sigmoid}(r_i)) \\
T_i &\sim \text{Bernoulli}(\text{sigmoid}(\alpha r_i + X_i^T \boldsymbol{\beta_\Delta})) .
\end{aligned}
\tag{6}
$$

We show results for two cases: when $\sigma^2$ is fixed and when $\alpha$ is fixed. Because this distribution of unobservables does not allow us to marginalize out $Z$, it converges more slowly than the uniform unobservables model and we must use a smaller sample size for computational tractability.

**Fixed $\sigma^2$:** We use the same simulation parameters as the uniform model. We fix $\sigma^2 = 2$ and we draw $\alpha \sim N(1, 0.1)$. We show results in Figure S2. Both the prevalence and expertise constraints produce more precise and accurate inferences for all parameters relative to the unconstrained model.

**Fixed $\alpha$:** We use the same simulation parameters as the uniform model, except we reduce the number of datapoints to 200. We fix $\alpha = 1$ and we draw $\sigma^2 \sim N(2, 0.1)$. We show results in Figure S3. Both the prevalence and expertise constraints produce more precise and accurate inferences for all parameters relative to the unconstrained model.

### D.4 MORE COMPLEX MODELS

To show our constraints are useful with more complex models, we ran two additional synthetic experiments on the Bernoulli-sigmoid model with uniform unobservables. First, we demonstrated applicability to higher-dimensional features. We show results in Figure S4. Even after quadrupling the number of features (which increases the runtime by a factor of three), both constraints still improve precision and accuracy. Secondly, we evaluate a more complex model with pairwise nonlinear interactions between features. We show results in Figure S5. Again both constraints generally improve precision and accuracy. We note our implementation relies on MCMC which is known to be less scalable than approaches like variational inference (Wainwright & Jordan, 2008) and would likely not scale to very high-dimensional features. However, our approach does not intrinsically rely

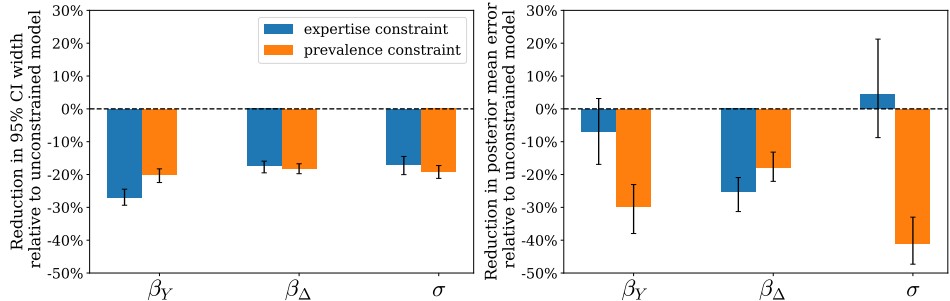

Figure S5: The prevalence and expertise constraints still improve parameter inference even when using pairwise nonlinear interactions between features (rather than only linear terms, as shown in Figure 2). Results are shown using synthetic data from the Bernoulli-sigmoid model with uniform unobservables. Both constraints generally produce more precise and accurate inferences on this synthetic data. We plot the median across 200 synthetic datasets. Errorbars denote the bootstrapped 95% confidence interval on the median.

on MCMC, and incorporating more scalable estimation methods is a natural direction for future work.[6]

# E  UK BIOBANK DATA

**Label processing:**   In the UK Biobank (UKBB), each person's data is collected at their baseline visit. The time period we study is the 10 years preceding each person's baseline visit. $T_i \in \{0, 1\}$ denotes whether the person receives a mammogram in the 10 year period. $Y_i \in \{0, 1\}$ denotes whether the person receives a breast cancer diagnosis in the 10 year period. We verify that very few people in the dataset have $T = 0$ and $Y = 1$ (i.e., are diagnosed with no record of a test): $p(Y = 1 | T = 0) = 0.0005$. We group these people with the untested $T = 0$ population, since they did not receive a breast cancer test.

**Feature processing:**   We include features which satisfy two desiderata. First, we use features that previous work has found to be predictive of breast cancer (NIH National Cancer Institute, 2017; Komen, 2023; Yanes et al., 2020). Second, since features are designed to be used in predicting $T_i$ and $Y_i$, they must be measured prior to $T_i$ and $Y_i$ (i.e., at the beginning of the 10 year study period). Since the start of our 10 year study period occurs before the date of data collection, we choose features that are either largely time invariant (e.g. polygenic risk score) or that can be recalculated at different points in time (e.g. age). The full list of features that we include is: breast cancer polygenic risk score, previous biopsy procedure (based on OPCS4 operation codes), age at first period (menarche), height, Townsend deprivation index[7], race (White, Black/mixed Black, and Asian/mixed Asian), and age at the beginning of the study period ($<$35, 35-39, and 40-45). We normalize all features to have mean 0 and standard deviation 1.

**Sample filtering:**   We filtered our sample based on four conditions. (i) We removed everyone without data on whether or not they received breast cancer testing, which automatically removed all men because UKBB does not have any recorded data on breast cancer tests for men. (ii) We removed everyone who was missing data (e.g. responded "do not know") for breast cancer polygenic risk score; previous biopsy procedure; menarche; height; Townsend deprivation index; race; age; duration of moderate physical activity; cooked, salad, and raw vegetable intake; weight; use of the following medication: aspirin, ibuprofen, celebrex, and naproxen; family history of breast cancer; and previous detection of carcinoma in breast. (iii) We removed everyone who did not self report being of White, Black/mixed Black, or Asian/mixed Asian race. (iv) We remove patients who were diagnosed with breast cancer before the start of our 10 year study period, as is standard in previous work (Zink et al., 2023). (v) We removed everyone above the age of 45 at the beginning of the observation period, since

---

[6]We use the same simulation parameters as our standard uniform model experiments. We set the expertise constraint to apply to a random subset of 60% of the features to match the standard uniform model experiments where expertise is assumed for 3 out of the 5 features.

[7]The Townsend deprivation index is a measure of material deprivation that incorporates unemployment, non-car ownership, non-home ownership, and household overcrowding (Townsend et al., 1988).

the purpose of our case study is to assess how the model performs in the presence of the distribution shift induced by the fact that young women tested for breast cancer are non-representative.[8]

**Model fitting:** We divide the data into train and test sets with a 70-30 split. We use the train set to fit our model. We use the test set to validate our risk predictions on the tested population ($T = 1$). We validate our risk predictions for the $T = 1$ population on a test set because the model is provided both $Y$ and $X$ for the train set, so using a test set replicates standard machine learning practice. We do not run the other validations (predicting risk among the $T = 0$ population and inference of unobservables) on a test set because in all these cases the target variable is unseen by the model during training. Overfitting concerns are minimal because we use a large dataset and few features.

**Inferred risk predicts breast cancer diagnoses among the untested population:** When verifying that inferred risk predicts future cancer diagnoses for the people who were untested ($T_i = 0$) at the baseline, we use data from the three UKBB follow-up visits. We only consider the subset of people who attended at least one of the follow-up visits. We mark a person as having a future breast cancer diagnosis if they report receiving a breast cancer diagnosis at a date after their baseline visit.

**Inferred unobservables correlate with known unobservables:** We verify that across people, our inferred posterior mean of unobservables correlates with a true unobservable—whether the person has a family history of breast cancer. We define a family history of breast cancer as either the person's mother or sisters having breast cancer. We do not include this data as a feature because we cannot be sure that the measurement of family history precedes the measurement of $T_i$ and $Y_i$. This allows us to hold out this feature as a validation.

**IRB:** Our institution's IRB determined that our research did not meet the regulatory definition of human subjects research. Therefore, no IRB approval or exemption was required.

## F  ADDITIONAL EXPERIMENTS ON CANCER DATA

Here we provide additional sets of experiments. We provide a comparison to various baseline models (Appendix F.1) and robustness experiments (Appendix F.2).

### F.1  COMPARISON TO BASELINE MODELS

We provide comparisons to three different types of baseline models: (i) a model trained solely on the tested population, (ii) a model which assumes the untested group is negative, and (iii) other selective labels baselines.

**Comparison to models trained solely on the tested population:** The first baseline that we consider is a model which estimates $p(Y_i = 1|T_i = 1, X_i)$: i.e., a model which predicts outcomes without unobservables using only the tested population.[9] This is a widely used approach in medicine and other selective labels settings. In medicine, it has been used to predict COVID-19 test results among people who were tested (Jehi et al., 2020; McDonald et al., 2021); to predict hypertrophic cardiomyopathy among people who received gold-standard imaging tests (Farahani et al., 2020); and to predict discharge outcomes among people deemed ready for ICU discharge (McWilliams et al., 2019). It has also been used in the settings of policing (Lakkaraju et al., 2017), government inspections (Laufer et al.), and lending (Björkegren & Grissen, 2020).

---

[8]To confirm that our predictive performance remains good when looking at patients of all ages, we conduct an additional analysis fitting our model on a dataset without the age filter, but keeping the other filters. (For computational tractability, we downsample this dataset to approximately match the size of the original age-filtered dataset.) We fit this dataset using the same model as that used in our main analyses, but add features to capture the additional age categories (the full list of age categories are: $<35$, 35-39, 40-44, 45-49, 50-54, $\geq55$). We find that if anything, predictive performance when using the full cohort is better than when using only the younger cohort from our main analyses in §5.2. Specifically, the model's quintile ratio is 4.6 among the tested population ($T_i = 1$) and 7.0 among the untested population ($T_i = 0$) that attended a follow-up visit.

[9]We estimate this using a logistic regression model, which is linear in the features. To confirm that non-linear methods yield similar results, we also fit random forest and gradient boosting classifiers. These methods achieve similar predictive performance to the linear model and they also predict an implausible age trend.

As shown in Figure S6, we find that the model trained solely on the tested population learns that cancer risk first increases with age and then falls sharply, contradicting prior epidemiological and physiological evidence (Komen, 2023; Cancer Research UK; US Cancer Statistics Working Group et al., 2013; Campisi, 2013). We see this same trend for a model fit without a prevalence constraint in §5.4. This indicates that these models do not predict plausible inferences consistent with prior work.

**Comparison to a model which treats the untested group as negative:** We also consider a baseline model which treats the untested group as negative; this is equivalent to predicting $p(T_i = 1, Y_i = 1|X_i)$, an approach used in prior selective labels work (Shen et al., 2021; Ko et al., 2020; Rastogi et al., 2023). We find that, though this baseline no longer learns an implausible age trend, it underperforms our model in terms of AUC (AUC is 0.60 on the tested population vs. 0.63 for our model; AUC is 0.60 on the untested population vs. 0.63 for our model) and quintile ratio (quintile ratio on the tested population is 2.4 vs. 3.3 for our model; quintile ratio for both models is 2.5 on the untested population). This baseline is a special case of our model with the prevalence constraint set to $p(Y = 1|T = 0) = 0$, an implausibly low prevalence constraint. In light of this, it makes sense that this baseline learns a more plausible age trend, but underperforms our model overall.

**Comparison to other selective labels baselines:** We also consider two other common selective labels baselines (Rastogi et al., 2023). First, we predict hard pseudo labels for the untested population (Lee, 2013): i.e., we train a classifier on the tested population and use its outputs as pseudo labels for the untested population. Due to the low prevalence of breast cancer in our dataset, the pseudo labels are all $Y_i = 0$, so this model is equivalent to treating the untested

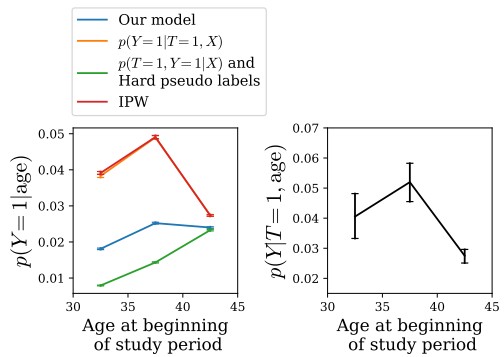

Figure S6: We run three sets of baseline models: (i) models trained solely on the tested population, estimating $p(Y_i = 1|T_i = 1, X_i)$; (ii) models which treat the untested group as negative, estimating $p(T_i = 1, Y_i = 1|X_i)$; and (iii) other selective labels baselines (IPW and hard pseudo labels). Both IPW and the model estimating $p(Y_i = 1|T_i = 1, X_i)$ learn that cancer risk first increases and then decreases with age, contradicting prior literature. This implausible inference occurs because the tested population has the same misleading age trend (right plot). In contrast, our Bayesian model learns a more plausible age trend (left plot, blue line). Hard pseudo labels and the model estimating $p(T_i = 1, Y_i = 1|X_i)$ also learn plausible age trends, but they underperform our Bayesian model in predictive performance.

group as negative and similarly underperforms our model in predictive performance. Second, we use inverse propensity weighting (IPW) (Shimodaira, 2000): i.e., we train a classifier on the tested population but reweight each sample by the inverse propensity weight $\frac{1}{p(T_i=1|X_i)}$.[10] As shown in Figure S6, this baseline also learns the implausible age trend that cancer risk first increases and then decreases with age: this is because merely reweighting the sample, without encoding that the untested patients are less likely to have cancer via a prevalence constraint, is insufficient to correct the misleading age trend.

### F.2 ROBUSTNESS CHECKS FOR THE BREAST CANCER CASE STUDY

Our primary breast cancer results (§5) are computed using the Bernoulli-sigmoid model in equation 4. In this model, unobservables are drawn from a uniform distribution, $\alpha$ is set to 1, and the prevalence constraint is set to $p(Y = 1) = 0.02$ based on previously reported breast cancer incidence statistics (Cancer Research UK). In order to assess the robustness of our results, we show that they remain consistent when altering all three of these aspects to plausible alternative specifications.

**Consistency across different distributions of unobservables:** We compare the uniform unobservables model (equation 4) to the normal unobservables model (equation 6). As described in Appendix

---

[10]We clip $p(T_i = 1|X_i)$ to be between [0.05, 0.95], consistent with previous work.

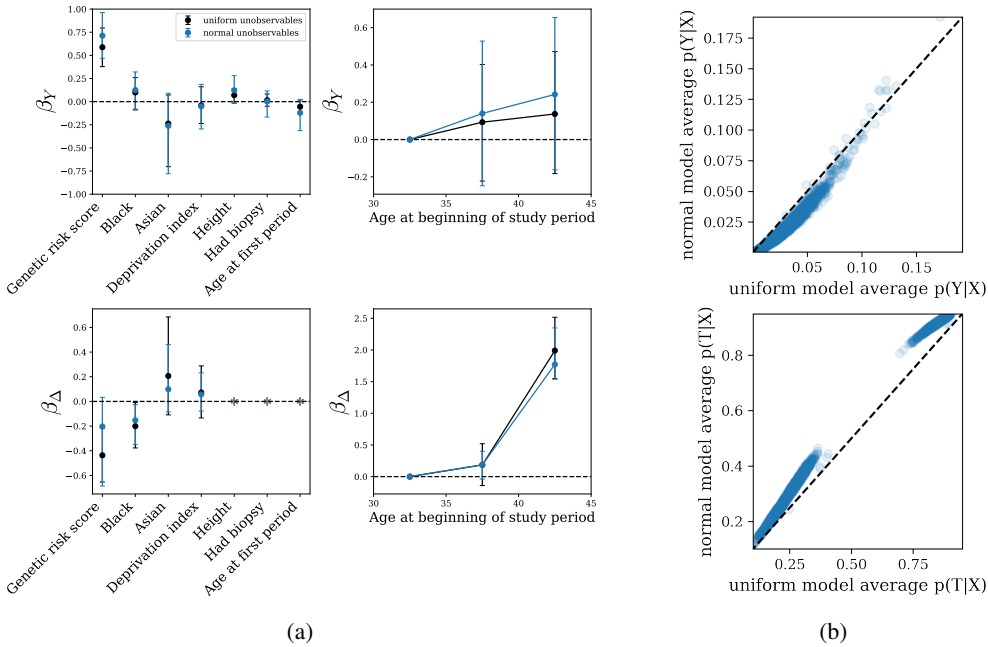

Figure S7: We compare the results from the uniform unobservable model in equation 4 (black) and the normal unobservable model in equation 6 (blue). Figure S7a: The estimated $\beta_Y$ and $\beta_\Delta$ coefficients remain similar for both models, with similar trends in the point estimates and overlapping confidence intervals. Figure S7b: Both models predict highly correlated values for $p(Y_i|X_i)$ and $p(T_i|X_i)$. Perfect correlation is represented by the dashed line.

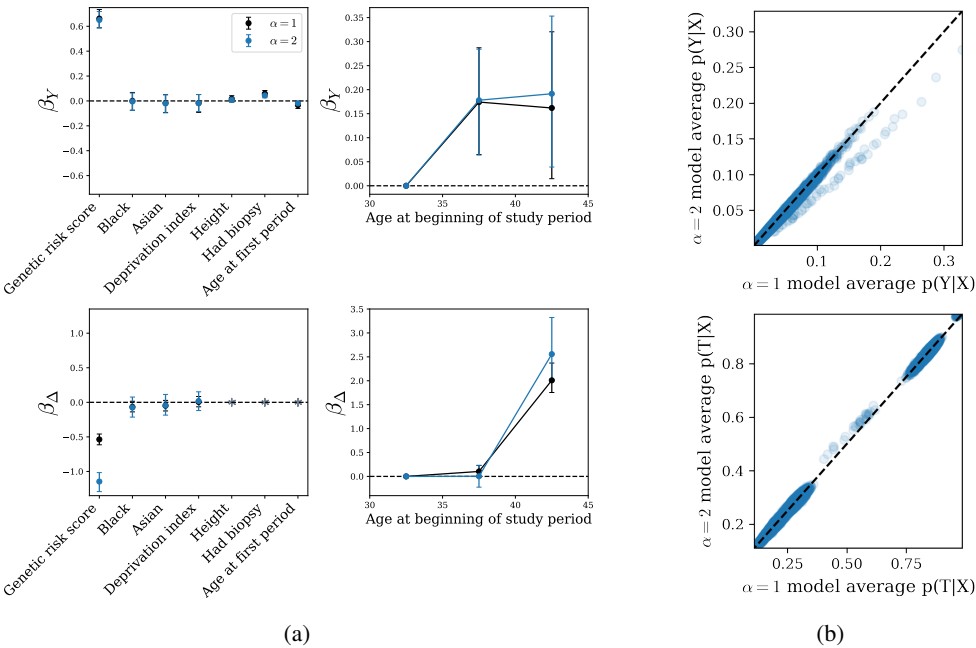

Figure S8: We compare the results from the uniform unobservable model with $\alpha = 1$ (black) and $\alpha = 2$ (blue). Figure S8a: The inferred $\beta_Y$ and $\beta_\Delta$ coefficients are generally very similar, with similar trends in the point estimates and overlapping confidence intervals. The only exception is the estimate of $\beta_\Delta$ for genetic risk, which is explained by the fact that the prediction of $\beta_\Delta$ depends on the value of $\alpha$. Figure S8b: Both models predict highly correlated values for $p(Y_i|X_i)$ and $p(T_i|X_i)$. Perfect correlation is represented by the dashed line.

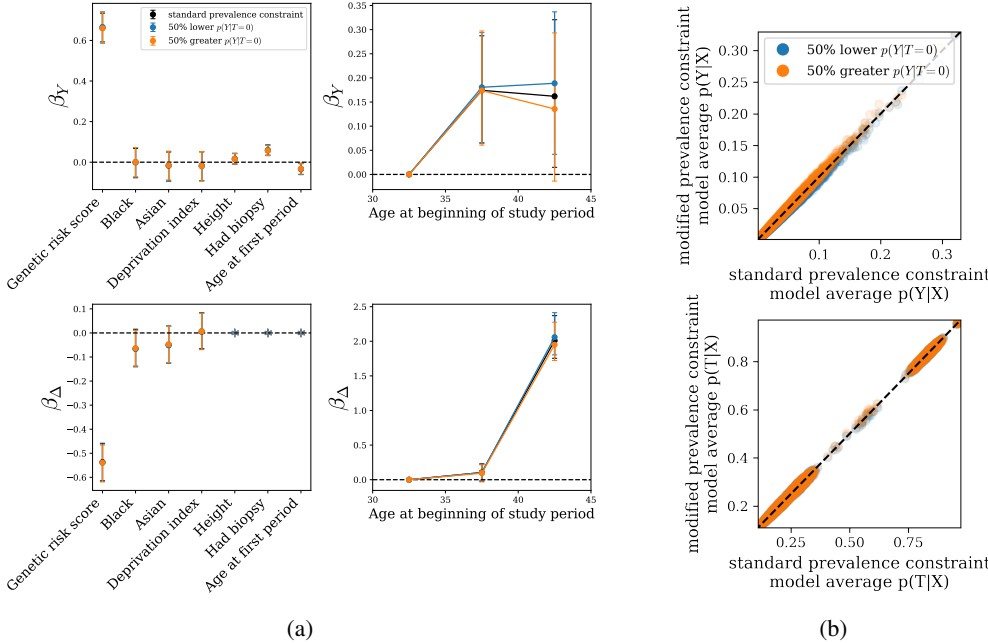

(a)                                                   (b)

Figure S9: We compare the results from the uniform unobservables model with a prevalence constraint of $\mathbb{E}[Y] = 0.02$ informed by cancer statistics (Cancer Research UK) (black), a prevalence constraint which corresponds to 50% less of the untested population having the disease (blue), and a prevalence constraint which corresponds to 50% more of the untested population having the disease (orange). Figure S9a: The predictions for all three models are similar as seen by the similar trends in the point estimates and overlapping confidence intervals. Figure S9b: All three models predict correlated values for $p(Y_i|X_i)$ and $p(T_i|X_i)$. Perfect correlation is represented by the dashed line.

D, the normal unobservables model does not allow us to marginalize out $Z_i$ and thus converges more slowly. Hence, for computational tractability, we run the model on a random subset of $\frac{1}{8}$ of the full dataset. In Figure S7a, we see that the estimated coefficients for both models remain similar, with similar trends in the point estimates and overlapping confidence intervals. Figure S7b shows that the inferred values of $p(Y_i|X_i)$ and $p(T_i|X_i)$ for each data point also remain correlated, indicating that the models infer similar testing probabilities and disease risks for each person.

**Consistency across different $\alpha$:** We compare the uniform unobservables model with $\alpha = 1$ to a uniform unobservables model with $\alpha = 2$. In Figure S8a, we see that the inferred coefficients for both models are generally very similar, with similar trends in the point estimates and overlapping confidence intervals. The only exception is $\beta_{\Delta}$ for the genetic risk score. While both models find a negative $\beta_{\Delta}$ for the genetic risk score, indicating genetic information is underused, the coefficient is less negative when $\alpha = 1$. This difference occurs because altering $\alpha$ changes the assumed relationship between the risk score and the testing probability under purely risk-based allocation, and thus changes the estimated deviations from this relationship (which $\beta_{\Delta}$ captures). Past work also makes assumptions about the relationship between risk and human decision-making (Pierson, 2020; Simoiu et al., 2017; Pierson et al., 2018; 2020). We can restrict the plausible values of $\alpha$, and thus $\beta_{\Delta}$, using the following approaches: (i) restricting $\alpha$ to a range of reasonable values based on domain knowledge; (ii) setting $\alpha$ to the value predicted by a model with $\sigma^2$ pinned; or (iii) fitting $\alpha$ and $\sigma^2$ in a model with non-binary $Y_i$ outcomes when both parameters can be simultaneously identified.

To confirm model consistency, we compare the inferred values of $p(Y_i|X_i)$ and $p(T_i|X_i)$ for each data point. As shown in Figure S8b, these estimates remain highly correlated across both models, indicating that the models infer similar testing probabilities and disease risks for each person.

**Consistency across different prevalence constraints:** The prevalence constraint fixes the estimate of $p(Y = 1)$. Because the proportion of tested individuals who have the disease, $p(Y = 1|T = 1)$, is known from the observed data, fixing $p(Y = 1)$ is equivalent to fixing the proportion of *untested* individuals with the disease, $p(Y = 1|T = 0)$. For the model in §5, we set the prevalence constraint to 0.02 based on cancer incidence statistics (Cancer Research UK). However, disease prevalence may not be exactly known (Manski & Molinari, 2021; Manski, 2020; Mullahy et al., 2021). To check the

robustness of our results to plausible variations in the prevalence constraint, we compare to two other prevalence constraints that correspond to 50% lower and 50% higher values of $p(Y = 1|T = 0)$.[11] This yields overall prevalence constraints of $\mathbb{E}[Y] \approx 0.018$ and $0.022$, respectively. In Figure S9a, we compare the $\boldsymbol{\beta_Y}$ and $\boldsymbol{\beta_\Delta}$ coefficients for these three different prevalence constraints. Across all three models, the estimated coefficients remain similar, with similar trends in the point estimates and overlapping confidence intervals. In particular, the age trends also remain similar in all three models, in contrast to the model fit without a prevalence constraint (§5.4). In Figure S9b, we compare the inferred values of $p(Y_i|X_i)$ and $p(T_i|X_i)$ for each data point and confirm that these estimates remain highly correlated across all three models, indicating that the models infer very similar testing probabilities and disease risks for each person.

---

[11]While our results are robust to significant alterations of the prevalence constraint, we do note that if the model is run with a wildly misspecified prevalence constraint — for example, $p(Y = 1|T = 0) = 0$ — it could produce incorrect results. To avoid this issue, our Bayesian framework also accommodates approximate constraints, if the prevalence is only approximately known.