# OpenReview forum: "Domain constraints improve risk prediction when outcome data is missing"
_ICLR.cc/2024/Conference — ICLR 2024 poster_

### Official Review · Reviewer_KKGr · 2023-10-31

**Soundness:** 4 excellent
**Presentation:** 4 excellent
**Contribution:** 4 excellent
**Rating:** 8
**Confidence:** 4

**Summary:**

This paper studies the selective labels problem applied to a healthcare context. Specifically, the paper proposes a Bayesian model for the problem and analyzes a special case of this model to show why two sensible constraints (a prevalence constraint, a human expertise constraint) improves inference. The paper also provides experimental results on synthetic and real data to show the effectiveness of the proposed model.

**Strengths:**

- The paper is very well-written and easy to follow
- The proposed model is simple and elegant; the authors do a good job explaining why the Heckman correction model is a special case
- The theoretical result is reassuring and also helps justify why the two suggested constraints help
- The experiments are well thought-out and I found the results to be compelling

**Weaknesses:**

I would like to see a more detailed discussion on how the model generalizes to more complex inputs (basically I'd like a more comprehensive discussion of Section 6's last sentence), especially as I think this is a very practically relevant extension. It would be helpful to understand to what extent the theory could explain this more complex setting (and under what assumptions one might need to additionally impose). It seems like a trivial extension would also be a partially linear model where some features are captured by a linear component and the rest are captured by a neural net.

Minor:
- Page 2: The text currently reads "Throughout, we generally refer to $Y_i$ as a binary indicator, but our framework extends to non-binary $Y_i$, and we derive our theoretical results in this setting" --- I would suggest rewording the last part so that it is clear what "this setting" refers to.

**Questions:**

See "weaknesses".

---

> ### Author Response · Authors · 2023-11-21
>
> We thank the reviewer for their positive review, and are glad that they found that our paper is well-written and easy to follow, that our model is elegant and simple, that the theoretical results justify why the constraints help parameter estimation, and that the experiments are well thought-out and compelling. We will now address the reviewer’s questions and suggestions for improvement.
>
> The reviewer first asks how the model generalizes to more complex inputs. To show our constraints are useful with more complex inputs, we ran two additional synthetic experiments. First, we demonstrated applicability to higher-dimensional features (Figure S4). Even after quadrupling the number of features (increasing runtime by a factor of three), both constraints still improve precision and accuracy. Secondly, we evaluate a more complex model with pairwise nonlinear interactions between features (Figure S5). Again both constraints generally improve precision and accuracy. We note our implementation relies on MCMC which is known to be less scalable than approaches like variational inference [1]. However, our approach does not intrinsically rely on MCMC (we are pursuing a follow-up paper investigating alternate approaches to fitting our models).
>
> Then the reviewer states, “A trivial extension would be a partially linear model where some features are captured by a linear component and the rest are captured by a neural net.” We thank the reviewer for this comment and agree that this is a natural direction for future work: for example, the neural net could make a prediction from a mammogram while the linear component incorporates clinical or demographic features. Another option is to use a purely linear model, but include features which are precomputed functions of more complex inputs. Indeed, our current model does this through the “genetic risk score” feature, capturing each patient's polygenic risk score which is a function of many genetic variants.
>
> Finally, the reviewer suggests that we reword the sentence: “Throughout, we generally refer to $Y_i$ as a binary indicator, but our framework extends to non-binary $Y_i$, and we derive our theoretical results in this setting” to make it more clear what “this setting” is. We thank the reviewer for this comment and will edit our manuscript to make this more clear. To clarify, we derive our theoretical results with the Heckman correction model which assumes a continuous $Y_i$. Some examples of non-binary $Y_i$ are tumor size or cancer stage.
>
> [1] Martin J Wainwright and Michael I Jordan. Graphical models, exponential families, and variational inference. *Foundations and Trends in Machine Learning*, 1(1–2):1–305, 2008.

---

### Official Review · Reviewer_6mUm · 2023-10-31

**Soundness:** 3 good
**Presentation:** 4 excellent
**Contribution:** 3 good
**Rating:** 8
**Confidence:** 4

**Summary:**

In this work, the authors introduce the widespread phenomenon that the data lies within the human decision censors that tend to be biased. The authors then proposed a hierarchical Bayesian model that addresses such data distribution mismatch between what has been tested and the underlying true distribution. The authors further proposed two constraints, prevalence constraint and expertise constraint to decrease the uncertainty of parameter estimation.

**Strengths:**

1. The proposed hierarchical Bayesian model to address the unobservables and connect it with the actual observation to evaluate the risk score and test decision makes sense and is novel.

2. The prevalence constraint and expertise constraint used to shrink the estimation uncertainty is novel. In practice, the two constraints are usually easy to access, making such constraints practically useful.

3. The authors demonstrated in synthetic data that the constraints proposed can effectively reduce the confidence interval and show in real data that the proposed constrained Bayesian model yields more reasonable discovery.

**Weaknesses:**

1. The actual Bayesian model derived from Proposition 3.1 seems too simple in practice. Having the assumption that the unobservable always comes from an independent normal distribution can be too strong.

2. When applying the model to UK Biobank, filtering out individuals whose age is below 45 is not convincing.

**Questions:**

Can you explain in more detail why, without prevalence constraint, the beta_y parameter will decrease when the age variable increases in Figure 4? You mentioned that being tested for breast cancer before age 50 is unusual, but that doesn't completely explain why you observe this trend.

---

> ### Author Response · Authors · 2023-11-21
>
> We thank the reviewer for their positive review and are glad that they found that our model makes sense and is novel, that our constraints are novel and practical, and that we verify the benefit of our constraints in both synthetic and real data.  We will now address the reviewer’s questions and suggestions for improvement.
>
> The reviewer first comments that the Heckman model introduced in Proposition 3.1 is too simple. We would like to clarify that our work investigates models *beyond* the Heckman model. In our paper we describe a broader class of models (described in equation 1), of which the Heckman model is only one special case (Proposition 3.1), and most of our experiments are run using models beyond the Heckman model. The reviewer also states, “Having the assumption that the unobservable always comes from an independent normal distribution can be too strong.” Indeed, this highlights a benefit of the general model class we describe, which works with alternate distributions of unobservables: in both our synthetic and real data experiments, we consider *both uniform and normal distributions of unobservables* (see Appendix C and Appendix E.2). Overall, we agree with the reviewer that the Heckman model is simple, and one of the strengths of our work is that we investigate models beyond the Heckman model.
>
> Then the reviewer states, “When applying the model to the UK Biobank, filtering out individuals whose age is below 45 is not convincing.” While we focus on the younger cohort in our main analyses to create a challenging distribution shift, to address the reviewer’s concern we run our model on the entire population. We find that performance when using the full cohort is *better* than when using the younger cohort. Specifically, AUC=0.67 and quintile ratio=4.6 among the tested population; AUC=0.66 and quintile ratio=7.0 among the untested population that attended a follow-up visit. We have added these results to footnote 6 in Appendix D.
>
> Finally, the reviewer asks why, for the model without the prevalence constraint, the $\beta_Y$ parameter decreases when the age variable increases in Figure 4. The model without the prevalence constraint learns this implausible age trend because it is learning the age trend which occurs among the tested population. Due to the age-based testing policy in the UK, patients under the age of 50 are tested only if they are of very high risk for breast cancer, so tested patients below the age of 50 have a higher risk than tested patients above the age of 50. The model without the prevalence constraint learns this trend, which is why its $\beta_Y$ decreases as age increases.

---

> > ### Comment · Reviewer_6mUm · 2023-11-22
> >
> > Thanks for the detailed clarification provided by the authors. The authors have addressed my questions and concerns. Therefore, I would like to increase the score.

---

### Official Review · Reviewer_26qi · 2023-11-01

**Soundness:** 3 good
**Presentation:** 4 excellent
**Contribution:** 3 good
**Rating:** 8
**Confidence:** 2

**Summary:**

The paper introduces a Bayesian model designed to infer risk and evaluate historical human decision-making in settings with selective labels. The authors integrate prevalence and expertise constraints, leading to enhanced parameter inference, as demonstrated both theoretically and empirically.

**Strengths:**

- The paper is well motivated.
- he constraints introduced are logical and reasonable.
- Both theoretical and empirical analysis show improved performance.

**Weaknesses:**

- The chosen Bernoulli-sigmoid model may be overly simplistic. Especially in the healthcare field, the intricate relationship between features and labels might not be fully represented by this basic model.
- The empirical tests were limited to only 7 features, raising questions about the model's scalability with a larger feature set.
- Section 5.2's results are somewhat ambiguous. For instance, in the subsection "Inferred risk predicts breast cancer diagnoses," it would be beneficial to include a specific predictive metric, such as the F1 score.
- The paper doesn't specify how the new model's diagnostic prediction performance stacks up against a model that doesn't factor in selective label issues. For instance, how would a straightforward linear model perform (1) by training solely on the tested population or (2) by treating the untested group as negative?

**Questions:**

- How does the model perform on the older population, where the distribution shift is less severe?
- Can you elaborate more on why the $\beta_{\Delta}$ is negative for genetic risk score?

---

> ### Author Response · Authors · 2023-11-21
>
> We thank the reviewer for their positive review that our draft was well-written. We now address their comments.
>
> The reviewer first comments that our model may be overly simplistic. To show our constraints are useful with more complex models, we ran two additional synthetic experiments. First, we demonstrated applicability to higher-dimensional features (Figure S4). Even after quadrupling the number of features (increasing runtime by a factor of three), both constraints still improve precision and accuracy. Secondly, we evaluate a more complex model with pairwise nonlinear interactions between features (Figure S5). Again both constraints generally improve precision and accuracy. We note our implementation relies on MCMC which is known to be less scalable than approaches like variational inference [1]. However, our approach does not intrinsically rely on MCMC (we are pursuing a follow-up paper investigating alternate approaches to fitting our models).
>
> The reviewer asks for specific predictive metrics for the results in section 5.2. To address this we report the AUC. (We report AUC instead of F1 score to allow comparison to past work.) The AUC amongst the tested population is 0.63 and amongst the untested population that attended a followup visit is 0.63. These AUCs are similar to past predictions which use similar feature sets [2] (and could be improved by using richer feature sets, though that is not the focus of this work). For instance, the Tyrer-Cuzick [3] and Gail [4] models achieved AUCs of 0.62 and 0.59.
>
> The reviewer asks how our model compares to models trained solely on the tested population. We fit a logistic regression model only on the tested population. (To confirm that non-linear methods did not yield an improvement, we also fit random forest and gradient boosted classifiers; this yielded very similar results to the logistic regression model.) These baselines suffer from the same issue: they learn that cancer risk first increases and then decreases with age which, as discussed in section 5.4, is implausible in light of prior research in oncology (Figure S6). For the tested population, our model achieves similar AUCs to these other models.
>
> The reviewer also asks how our model compares to treating the untested group as negative; this is equivalent to predicting $p(T=1, Y=1|X)$, an approach used in prior work [5,6]. Though this baseline no longer learns an implausible age trend, it underperforms our model in AUC (for both the tested and untested population AUC is 0.60 vs. 0.63 for our model) and quintile ratio (quintile ratio on the tested population is 2.4 vs. 3.3 for our model; quintile ratio on the untested population is 2.5 for both models). This baseline is a special case of our model with an implausibly low prevalence constraint $p(Y=1|T=0) = 0$. In light of this, it makes sense that this baseline learns a plausible age trend, but underperforms our model overall.
>
> Though the reviewer did not explicitly request this, we also compare to two other selective labels baselines. First, we predict hard pseudo labels [7]: i.e., we train a classifier on the tested population and use its outputs as pseudo labels for the untested population. Due to the low prevalence of cancer, the pseudo labels are all $Y=0$, so this model is equivalent to treating the untested group as negative and similarly underperforms our model. Second, we use inverse propensity weighting [8]: i.e., we train a classifier on the tested population but reweight each sample by $\frac{1}{P(T=1|X)}$. This baseline learns the implausible age trend because it merely reweights the sample, without encoding that the untested patients are less likely to have cancer via a prevalence constraint. All of the baseline results have been added to Appendix E.1.
>
> The reviewer asks, “How does the model perform on the older population?” To address this we run our model on the entire population. We find that performance when using the full cohort is *better* than when using the younger cohort. Specifically, AUC=0.67 and quintile ratio=4.6 among the tested population; AUC=0.66 and quintile ratio=7.0 among the untested population that attended a follow-up visit. We also evaluate this model using only adults over 50 (as opposed to the entire cohort) and performance remains better than our initial analysis.  Specifically, AUC=0.67 and quintile ratio=5.1 among the tested population; AUC=0.80 and the quintile ratio is infinite (4.5% vs. 0%) among the untested population. (Performance for the untested population is noisily estimated because there are relatively few untested adults over 50 due to the testing guideline.) We have added these results to footnote 6 in Appendix D.
>
> Finally, the reviewer asks “why is $\beta_\Delta$ negative for genetic risk score?” A negative $\beta_\Delta$ indicates that, controlling for cancer risk, patients with high genetic risk are under-tested. This is plausible because doctors frequently lack patient genetic information.

---

> > ### Author Response · Authors · 2023-11-21
> >
> > [1] Martin J Wainwright and Michael I Jordan. Graphical models, exponential families, and variational inference. *Foundations and Trends in Machine Learning*, 1(1–2):1–305, 2008.
> >
> > [2] Adam Yala, Peter G Mikhael, Fredrik Strand, Gigin Lin, Kevin Smith, Yung-Liang Wan, Leslie Lamb, Kevin Hughes, Constance Lehman, and Regina Barzilay. Toward robust mammography-based models for breast cancer risk. *Science Translational Medicine*, 13(578), 2021.
> >
> > [3] Jonathan Tyrer, Stephen W Duffy, Jack Cuzick, A breast cancer prediction model incorporating familial and personal risk factors. *Stat. Med*. 23:1111–1130, 2004.
> >
> > [4] Mitchell H Gail, Louise A Brinton, David P Byar, Donald K Corle, Sylvan B Green, Catherine Schairer, John J Mulvihill, Projecting individualized probabilities of developing breast cancer for white females who are being examined annually. *J. Natl. Cancer Inst*. 81:1879–1886, 1989.
> >
> > [5] Yiqiu Shen, Farah E Shamout, Jamie R Oliver, Jan Witowski, Kawshik Kannan, Jungkyu Park, Nan Wu, Connor Huddleston, Stacey Wolfson, Alexandra Millet, et al. Artificial intelligence system reduces false positive findings in the interpretation of breast ultrasound exams. *Nature Communications*, 12(1):1–13, 2021.
> >
> > [6] Wei-Yin Ko, Konstantinos C Siontis, Zachi I Attia, Rickey E Carter, Suraj Kapa, Steve R Ommen, Steven J Demuth, Michael J Ackerman, Bernard J Gersh, Adelaide M Arruda-Olson, et al. Detection of hypertrophic cardiomyopathy using a convolutional neural network-enabled electrocardiogram. *Journal of the American College of Cardiology*, 75(7):722–733, 2020.
> >
> > [7] Dong-Hyun Lee. Pseudo-label : The simple and efficient semi-supervised learning method for deep neural networks. *International Conference of Machine Learning 2013 Workshop: Challenges in Representation Learning (WREPL)*, 2013.
> >
> > [8] Hidetoshi Shimodaira. Improving predictive inference under covariate shift by weighting the log-likelihood function. *Journal of Statistical Planning and Inference*, 90(2):227–244, 2000.

---

> > > ### Comment · Reviewer_26qi · 2023-11-23
> > >
> > > Thanks to the authors for their reply. I will increase my ratings.

---

### Official Review · Reviewer_uugS · 2023-11-05

**Soundness:** 2 fair
**Presentation:** 3 good
**Contribution:** 2 fair
**Rating:** 5
**Confidence:** 3

**Summary:**

The paper proposes a Bayesian model for disease risk of the patients where only the outcome of the tested patients are observed. The proposed model has linear model for risk and testing decision on the observed variables. The paper introduces two constraints: prevalence constraint -- sets expectation of outcome based on prevalence of the disease and expertise constraint --  fixes some parameters to zero based on domain knowledge . The proposed approach is tested in a synthetic and real breast cancer data.

**Strengths:**

1. The paper is very well-written, readers can easily follow the motivation, problem formulation and their experimental design.
2. I appreciate the experiments trying to run experiments in real breast cancer dataset. The experiments in a setting where outcomes for non-tested patients are missing is a very difficult setting.
3. The paper addresses a significant problem where the outcomes of the patients that are tested are missing and there is distributional shift between tested and untested patients. There is a variety of applications -- which are also motivated in the paper.

**Weaknesses:**

1. I think the paper has limited novelty. The linear risk setting has been considered before as cited in the paper before [(Hicks, 2021)]. This paper aims to add two more constraints: prevalence constraint and expertise constraint. The expertise constraint sets one of the variables to 0 - could be easily addressed by dropping that feature in the dataset, and prevalence constraint sets the expectation of the outcome -- could be addressed by normalizing the feature space and adding a bias term. I am not convinced that these contributions are significant enough to grant acceptance.
2. I am not sure what theoretical results bring in the paper. For example, Proposition 3.2 shows that variance on the unknown parameters are less if you condition on the fixed parameters. Isn't this expected ? I am not sure how much value this adds to the paper.

**Questions:**

1. The experimental setting for breast cancer patients are interesting -- you are using patient follow-up to validate the methodology ? What happens if there is no follow-up ? How accurate is it to use follow-up data ?

---

> ### Author Response · Authors · 2023-11-21
>
> We are glad the reviewer found that the paper was well-written and easy to follow, that we applied our model to real data from a difficult medical setting, and that we addressed a significant distribution shift problem which is applicable to many domains. We will now address the reviewer’s questions and suggestions for improvement.
>
> The reviewer first comments about the novelty of our work, stating “the linear risk setting has been considered before as cited in [(Hicks, 2021)].” We clarify that the novelty of our work is twofold. First, (Hicks, 2021) only considers the Heckman correction model; in our paper we describe a *broader* class of models (described in equation 1), of which the Heckman correction model is only one special case (Proposition 3.1). Secondly, we propose two novel constraints — the prevalence and expertise constraints — which are *not* considered in the Heckman model. These constraints are straightforward to implement and well-motivated in a medical setting. We validate both empirically and theoretically that these constraints improve parameter estimation.
>
> The reviewer also provides individual comments for each of our constraints. For the expertise constraint, the reviewer states, “The expertise constraint sets one of the variables to 0 - this could be easily addressed by dropping that feature in the dataset.” We clarify that the expertise constraint only drops a subset of features when predicting the *testing decision*. Thus, $\beta_\Delta$ for these features are set to 0. However, these features are *not* dropped when predicting *disease risk*. Thus we still estimate $\beta_Y$ for all features (even for the features whose $\beta_\Delta$ is set to 0). Therefore the features for which we assume expertise *cannot* be completely dropped from the dataset. For the prevalence constraint, the reviewer states, “the prevalence constraint sets the expectation of the outcome - this could be addressed by normalizing the feature space and adding a bias term.” While this is correct for a linear model with an identity link, this is *not* true in general (e.g. for a Bernoulli-sigmoid link). Overall, we thank the reviewer for their comments on both constraints, and we will revise the manuscript to clarify these points. We agree with the reviewer that the constraints are straightforward to implement: this is a benefit which makes them compatible with a wide class of models.
>
> The reviewer also asks what value our theoretical results, such as Proposition 3.2, add to the paper. The significance of the theoretical results is twofold. First, we show that the Heckman model is a special case of our model, providing an important connection to the econometrics literature and providing intuition about model identifiability. Secondly, we provide conditions under which fixing a parameter reduces the variance of the other parameters, improving the precision of parameter inference. While Proposition 3.2 is general, we also provide more specific results for the Heckman model (Proposition A.2). Ultimately, our theoretical results (i) provide a connection to the Heckman model, showing it is a special case of our general model and (ii) provide an explanation for why our constraints improve parameter estimation.
>
> Finally, the reviewer asks what happens if we do not have follow-up data for certain patients. We use follow-up data to validate that our model’s inferred risk predictions indeed predict future breast cancer diagnoses *even among the untested population*, an approach also leveraged by prior work [1]. This is an improvement on merely assessing the model on the tested population, since it allows us to get some sense of whether we are able to accurately predict risk for the untested population, though the reviewer is correct that we are not guaranteed to have follow-up data for all untested patients (and we exclude patients with missing follow-up data from this analysis). To get around this limitation, we conduct three additional validations in section 5.2. Nevertheless, we thank the reviewer for their comment and we will acknowledge the limitations of this validation in the main text.
>
> [1] Sendhil Mullainathan and Ziad Obermeyer. Diagnosing physician error: A machine learning approach to low-value health care. *The Quarterly Journal of Economics*, 137(2):679–727, 2022

---

### Author Response · Authors · 2023-11-21

We appreciate the thoughtful comments from all four reviewers and are glad that the reviews are overall positive. We thank the reviewers for noting that our paper was well-written, our constraints are reasonable and practically useful, and that our theoretical results justify why our two constraints help model estimation, among other positive comments. We have thoroughly revised and updated our manuscript to incorporate the reviewers’ comments. We summarize the major changes here and separately provide detailed responses to each reviewer individually. Based on the reviewer comments, we have conducted three additional sets of experiments, all of which are now added to the manuscript.

1. We run experiments with more features and with more complex models. In particular, we run synthetic experiments with quadruple the number of features and find that even in this setting the constraints improve precision and accuracy (Figure S4). We also run synthetic data experiments on a more complex model with nonlinear interactions between features. Here as well we find that the constraints improve precision and accuracy (Figure S5).
2. We compare our model’s performance to a suite of additional baselines (Appendix E.1). This includes (i) baselines trained solely on the tested population, (ii) baselines which treat the untested population as negative, and (iii) additional baselines commonly used in selective labels settings. Collectively, these baselines all suffer from various issues our model does not, including learning implausible age trends inconsistent with prior literature or worsening predictive performance (Figure S6).
3. Based on reviewer questions about how the model performs on the entire cohort, as opposed to the age-filtered cohort, we rerun our breast cancer case study without applying the age filter. The model’s predictive performance, if anything, is better when using the entire cohort (footnote 6 in Appendix D).

---

### Meta-Review · Area_Chair_8NzX · 2023-12-10

**Metareview:**

**Strengths**

- Clearly written paper
- Addresses important problem in medicine and potentially other applications.

**Weaknesses / What's Missing**

(**Minor**) Oversight of "Learning from One-Sided Feedback: This work overlooks substantial literature on "learning when outcome data are missing." The term to search for is: "learning from one-sided feedback." A good entry point [1]. I would recommend the authors go through this paper and the papers referenced therein, citing as appropriate. The "selective labels" paper by Lakkaraju et al. inadvertently rediscovered this problem. It would be helpful to remedy this oversight for future readers.

(**Minor**) Potential Solution for Censoring and Bias Effects: The paper may have two other applications beyond the ones that are mentioned here. One of them is to potentially address the "censoring" discussed in Chien et al. [2], . Another is to potentially address the bias issue that are discussed in [3]. Both of these would require appropriate domain constraints, but are worth mentioning since they suffer from some of the key limitations mentioned in the Introduction. In this vein, the earlier work on learning from one-sided feedback may also reveal some other potential applications.

(**Major**) Availability of Correct Domain Constraints: The work broadly assumes that the information required to formulate "domain constraints" is broadly available in medical applications. The authors do a good job in pointing to work that provides standalone estimates of, e.g., prevalence. What is missing is:

(i) a discussion on whether these estimates are reliable;
(ii) a description of how to estimate these quantities from scratch, or check their reliability;
(iii) a discussion of how the proposed approach will perform if these quantities are misspecified (Major)

To put things in perspective, this is the weakest part of the submission and would benefit greatly from revision and development in a potential camera-ready. I am recommending acceptance at this stage since all of these issues can be addressed and the contributions will not depend on the outcome itself. In particular, (i) and (ii) simply require a more thoughtful discussion. Issue (iii) requires a small simulation study to determine how results change under misspecification, but the outcome of the study would not alter the overall constraints.

-----

**References**

[1] Jiang et al. [Learning the Truth From Only One Side of the Story](https://proceedings.mlr.press/v130/jiang21b/jiang21b.pdf), AISTATS 2021

[2] Chien et al. [Algorithmic Censoring in Dynamic Learning Systems](https://dl.acm.org/doi/10.1145/3617694.3623247), EAAMO 2023

[3] Jia et al. [Anthropogenic Biases in Chemical Reaction Data Hinder Exploratory Inorganic Synthesis](https://www.nature.com/articles/s41586-019-1540-5), Nature 2019

**Justification For Why Not Higher Score:**

- Low confidence reviews
- Weaknesses listed above
- Paper reads like a class project at times

**Justification For Why Not Lower Score:**

The paper tackles a common problem that arises in a large class of important applications. The proposed solution is sound, viable, and potentially generalizable to other settings.

---

### Decision · Program_Chairs · 2024-01-16

Accept (poster)